# HUMANOIDOLYMPICS: SPORTS ENVIRONMENTS FOR PHYSICALLY SIMULATED HUMANOIDS

## ABSTRACT

We present HumanoidOlympics, a collection of physically simulated sports environments designed for the animation and robotics communities to develop humanoid behaviors. Our suite includes individual sports such as golf, javelin throw, high jump, long jump, and hurdling, as well as competitive games like table tennis, tennis, fencing, boxing, soccer, and basketball. By simulating a wide range of Olympic sports, HumanoidOlympics offers a rich and standardized testing ground to evaluate and develop learning algorithms due to the diversity and physically demanding nature of athletic activities. Our suite supports simulating both graphics-focused (SMPL and SMPL-X) and real-world humanoid robots. For each sport, we benchmark popular humanoid control methods and provide expert-designed rewards that lead to surprising simulation results. Our analysis shows that leveraging human demonstrations can significantly enhance the resulting policies' human likeness and task performance. By providing a unified and competitive sports benchmark, HumanoidOlympics can help the animation and robotics communities develop human-like and performant controllers. Supplementary videos are available at https://humanoidolympics.github.io/.

## 1 INTRODUCTION

Competitive sports, much like their role in human society, offer a standardized way of measuring the performance of learning algorithms and creating emergent behavior. Natural and human-like humanoid motion in sports can be used in animation, robotics motion planning, and more, where the quality of the motion is important. Human-like motion is not only visually appealing but also serves as a functional and efficient foundation for various tasks such as navigation, grasping, etc.. While there exist isolated efforts to bring individual sport into physics simulation [12, 50, 11, 49, 43, 39], each work uses a different humanoid, simulator, and learning algorithm, which prevents unified evaluation. Their specially built humanoids also make it difficult to acquire compatible motion data. Building a collection of simulated sports environments that supports a range of standardized humanoid embodiment and training pipeline is challenging, as it requires expert knowledge in humanoid control, reinforcement learning (RL), and physics simulation.

These challenges have led to previous benchmarks and simulated environments [2, 35] focusing mainly on locomotion tasks for humanoids. While these tasks (e.g., moving forward, getting up from the ground, traversing terrains) are benchmarks, they lack the depth and diversity needed to induce a wide range of behaviors and strategies. As a result, these environments do not fully exploit the potential of humanoids to discover actions and skills found in real-world human activities.

Another important challenge of working with simulated humanoids is the ease of obtaining human demonstrations. The resemblance to the human body makes humanoids capable of performing a diverse set of skills; a human can also easily judge the strategies used by humanoids. Curated human motion can be used either as motion prior [23, 24, 33] or in evaluation protocols. Thus, having an easy way to obtain new human motion data compatible with the humanoid, either from motion capture (MoCap) or videos, is critical for simulated humanoid environments.

In this work, we propose HumanoidOlympics, a collection of physically simulated environments for a variety of Olympic sports. Tackling these environments requires not only locomotion skills, but also manipulation, coordination, and planning. Our environments also support multiple humanoid embodiments and provide a rich set of challenges for developing and testing embodied agents. We

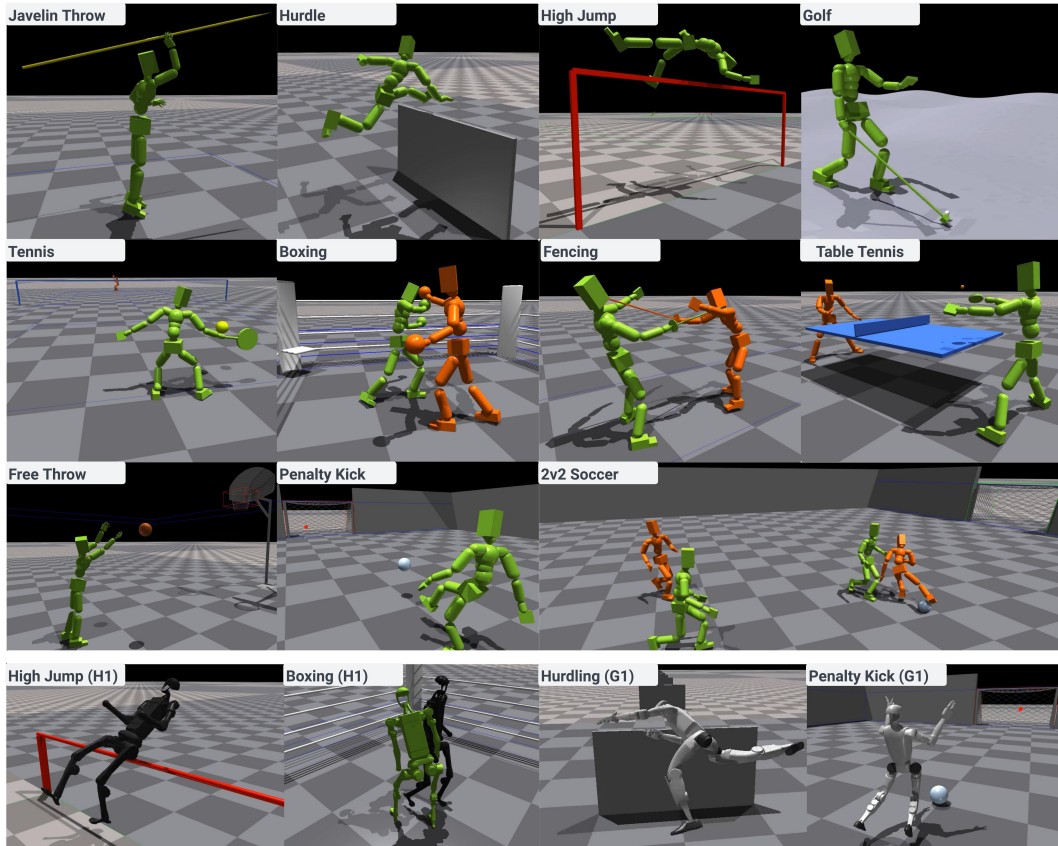

Figure 1: A collection of various sports environments for physically simulated humanoids. Top three rows: SMPL and SMPL-X humanoid. Bottom row: Unitree humanoids H1 and G1.

use humanoids compatible with the SMPL [13, 22] family of human models and humanoid robots such as Unitree H1 [37] and G1 [36]. As popular human models, the SMPL family of models is widely adopted in the vision and graphics community, which provides us with access to human pose estimation methods [48] capable of extracting coherent motion from videos. The existing large-scale human motion dataset [18] in the SMPL format also helps build general-purpose motion representation for humanoids [16]. From motions described in the SMPL format, we can retarget them to real-world humanoids [9, 8, 5, 3]. We conduct most of our quantitative experiments using the SMPL family humanoids, and have qualitative results on all the humanoid robots.

Our sports environments support both individual and competitive sports: for individual sports, we include golf, javelin throw, high jump, long jump, hurdling, and free throws; competitive sports in our suite include 1v1 games such as ping pong, tennis, fencing, and boxing, as well as team sports such as soccer. We also define tasks such as penalty kicks (for soccer) and ball-target hitting (for ping-pong and tennis) that are easier to define performance metrics and can be used in curriculum learning.

To demonstrate the importance of human demonstrations, we test recently proposed humanoid motion representations learned from large-scale human Motion Capture (MoCap) [16], and show that a strong motion prior combined with simple rewards can lead to many versatile human-like behaviors to achieve impressive sports results (*i.e.* discovering the Fosbury way for high jump). For sports that has no MoCap, we extract motion from videos using off-the-shelf pose estimation methods, and show that using human motion data as adversarial motion prior [24] can significantly improve human likeness in the resulting motion.

In conclusion, our contributions are: (1) we propose HumanoidOlympics, a collection of simulated sports environments that support multiple humanoid embodiments (SMPL, SMPL-X, Unitree H1 and G1); (2) for each sport, we provide example state and reward designs, benchmark state-of-the-art algorithms, and show that carefully designed rewards combined with a strong motion prior can lead to

impressive sports feats and novel simulation results; (3) we provide a pipeline to extract task-specific human demonstration data from videos and show their effectiveness in helping build human-like strategies in simulated sports.

## 2 RELATED WORKS

**Simulated Humanoid Sports**. Simulated humanoid sports can help generate animations and explore optimal sports strategies. Research has focused on various individual sports within simulated environments, including tennis [50], table tennis [39], boxing [43, 52], fencing [43], basketball [11, 40, 41] and soccer [45, 12]. These studies leverage human demonstration to achieve human-like behaviors, using it to acquire motor skills [12, 43] or establish motion prior [50]. However, the diverse and non-standard humanoid models across studies makes it difficult to aggregate additional human demonstration data. Furthermore, the task-specific training pipelines in these studies are hard to generalize to new sports. In contrast, HumanoidOlympics provides a unified benchmark employing a standard humanoids and training pipeline across all sports. This standardization not only facilitates extension to more sports, but also simplifies benchmarking learning algorithms.

**Simulated RL Benchmarks**. Simulated full-body humanoids provide a valuable platform for studying embodied intelligence due to their close resemblance to real-world human behavior and physical interactions. Current RL benchmarks [2, 35, 19] often focus on locomotion tasks such as moving forward and traversing terrain. `dm_control` [35] and OpenAI [2] Gym focus exclusively on locomotion tasks. ASE [25] includes results for five tasks based on mocap data, which involve mainly locomotion and sword-swinging actions. These tasks lack the complexity required to fully exploit the capabilities of simulated humanoids. Sports scenarios require agile motion and strategic teamwork. They are also easily interpretable and provide measurable outcomes for success. A concurrent work, HumanoidBench [30] employs the Unitree H1 humanoid in simulation to address 27 locomotion and manipulation tasks. Unlike HumanoidBench, ours targets competitive sports and uses available human demonstration data to enhance the learning of human-like behaviors. This emphasis is essential, as without human demonstrations, the behaviors developed in benchmarks can often appear erratic and lack human realism.

**Humanoid Motion Representation**. Due to the high degree-of-freedom (DoF) in humanoids and the inherent sample inefficiency of RL training, there have been many efforts focusing on developing motion primitives [7, 20, 6, 27] and motion latent spaces [4, 25, 33]. These techniques aim to accelerate training and provide human-like motion priors. Notably, approaches such as ASE [25], CASE [4], PMP [1], CompositeMotion [46], and CALM [33] utilize adversarial learning objectives to encourage mapping between random noise and realistic motor behavior. Furthermore, methods such as ControlVAE [47], NPMP [20], PhysicsVAE [44], NCP [52], MaskedMimic [34], and PULSE [16] leverage the motion imitation task to acquire and reuse motor skills for the learning of downstream tasks. In this work, we study AMP [24] and PULSE [16] as exemplary methods to provide motion priors. Our findings demonstrate that a robust motion prior, combined with straightforward reward designs, can effectively induce human-like behaviors in solving complex sports tasks.

## 3 PROBLEM FORMULATION

We define the full-body human pose as $q_t \triangleq (\theta_t, p_t)$, consisting of 3D joint rotations $\theta_t \in \mathbb{R}^{J \times 6}$ and positions $p_t \in \mathbb{R}^{J \times 3}$ of all $J$ joints on the humanoid, using the 6 DoF rotation representation [51]. To define velocities $\dot{q}_{1:T}$, we have $\dot{q}_t \triangleq (\omega_t, v_t)$ as angular $\omega_t \in \mathbb{R}^{J \times 3}$ and linear velocities $v_t \in \mathbb{R}^{J \times 3}$. If an object is involved (*e.g.* javelin, football, ping-pong ball), we define their 3D trajectories $q_t^{\text{obj}}$ using object position $p_t^{\text{obj}}$, orientation $\theta_t^{\text{obj}}$, linear velocity $v_t^{\text{obj}}$, and angular velocity $\omega_t^{\text{obj}}$. As a notation convention, we use $\widehat{\cdot}$ to denote the ground truth kinematic quantities from MoCap and normal symbols without accents for values from the physics simulation.

**Goal-conditioned Reinforcement Learning for Humanoid Control**. We define each sport using the general framework of goal-conditioned RL. Namely, a goal-conditioned policy $\pi_{\text{task}}$ is trained to control a simulated humanoid competing in a sports environment. The learning task is formulated as a Markov Decision Process [26, MDP] defined by the tuple $\mathcal{M} = \langle \mathcal{S}, \mathcal{A}, \mathcal{T}, \mathcal{R}, \gamma \rangle$ of states, actions,

transition dynamics, reward function, and discount factor. The simulation determines the state $s_t \in \mathcal{S}$ and transition dynamics $\mathcal{T}$, where a policy computes the action $a_t$. The state $s_t$ contains the proprioception $s_t^{\mathrm{p}}$ and the task observation / goal state $s_t^{\mathrm{g}}$. Proprioception is defined as $s_t^{\mathrm{p}} \triangleq (q_t, \dot{q}_t)$, which contains the 3D body pose $q_t$ and velocity $\dot{q}_t$. We use $b$ to indicate the boundary of the arena to which a sport is limited. All values are normalized with respect to the humanoid heading (yaw).

**Humanoid Embodiment**. We support four types of humanoids, shown in Fig. 2. The SMPL humanoid models adhere to the SMPL [13] kinematic structure, featuring 24 joints, 23 of which are actuated, yielding an action space of $\mathcal{R}^{69}$. The SMPL-X [22] humanoid has 52 joints, 51 actuated, including 21 body joints and hands, resulting in an action space of $\mathcal{R}^{153}$. Body parts on our humanoid consist of primitives such as capsules and blocks. Since the SMPL humanoid is used for animation purposes, we impose a 500Nm torque limit on the humanoid and

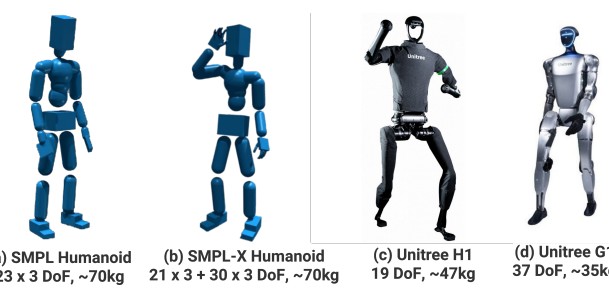

(a) SMPL Humanoid
23 x 3 DoF, ~70kg

(b) SMPL-X Humanoid
21 x 3 + 30 x 3 DoF, ~70kg

(c) Unitree H1
19 DoF, ~47kg

(d) Unitree G1
37 DoF, ~35kg

Figure 2: Supported Humanoid Embodiments. All humanoids have a similar maximum torque to weight ratio.

run simulation in 60 Hz and control in 30 Hz, following prior art [24]. For real-world humanoids, we support the full-sized H1 [37] (19 DoF) and the smaller G1 [36] (37 DoF). To simulate the H1 and G1 robots, we adopt the simulation parameters (e.g., 200 Hz simulation and 50 Hz control, joint and torque limits) used in prior art for humanoid sim-to-real transfer [9]. However, as our focus is on simulation, we do not conduct domain randomization and sim-to-real adaptation.

## 4 SPORTS ENVIRONMENTS FOR SIMULATED HUMANOIDS

In this section, we introduce our diverse suite of sports environments. An overview can be found in Fig. 3. Each environment is designed with a scoring system and success metrics that closely mirror real-world sports performance evaluation. These metrics enable proper comparison between methods. While important for comparisons, these metrics are very sparse. For example, a high jump is evaluated at the end of the run-then-jump sequence. This form of sparse feedback is detrimental to RL algorithms, leading to the well-known credit assignment problem [21]. To overcome this, we design dense reward functions to guide the learning process.

Our environments support both single-person (Sec. 4.1) and multi-person (Sec. B.2) sports. For each environment, we provide the observation space and the key performance metrics used for evaluation. While reward engineering is crucial for effective learning, we will focus on two exemplary environments in depth here: one for single-person sports (golf), and one for multi-person sports (soccer). Full details for all environments in Appendix B.1.

We leverage human demonstration in two ways: (1) we show a motion representation/prior trained on large-scale MoCap data can help learn many sports tasks results in human-like behavior; (2) we propose a pipeline (Sec. 4.3) to obtain a small amount of task-specific demonstration from video for each sport (when there is no available public dataset), and show that it can further help shaping the humanoid behavior when used together with a general-purpose motion prior.

### 4.1 SINGLE-PERSON SPORTS

**Golf**. In our golf environment, the objective is to hit a ball into a hole using a club. We modify the humanoid model by replacing its right hand with a 1.4-meter-long golf club. The club's end (driver) is simulated as a small box (0.05m × 0.025m × 0.02m). To simulate realistic golfing terrain, we generate a wave-like ground mesh with a 0.5-meter amplitude. Each episode randomly positions the hole (goal) 0 to 20 meters to the left of the humanoid.

The task observation $s_t^{\mathrm{g\text{-}golf}} \triangleq (p_t^b, p_t^c, p_t^g, o_t)$ includes the ball position $p_t^b \in \mathbb{R}^3$, club $p_t^c \in \mathbb{R}^3$, goal position $p_t^g \in \mathbb{R}^3$, and terrain height map $o_t \in \mathbb{R}^{32 \times 32}$.

Figure 3: An overview of HumanoidOlympics: we design a collection of simulated sports environments and leverage RL and human demonstrations (from videos or MoCap) as prior to tackle them.

Performance metrics in this task are sparse and evaluated only when the ball stops moving. Success is determined by whether the ball enters the hole, and we also measure the distance between the hole and the ball's final position. To address the challenges posed by such long-horizon and sparse rewards for RL algorithms, we implement a dense reward function comprising several components to guide learning:

$$\mathcal{R}^{\text{golf}}(\boldsymbol{s}_t^{\text{p}}, \boldsymbol{s}_t^{\text{g-golf}}) \triangleq 1 \times r_t^{\text{p}} + 1 \times r_t^{\text{c}} + 1 \times r_t^{\text{g}} + 1 \times r_t^{\text{pred}} \tag{1}$$

The position reward, $r_t^{\text{p}} \triangleq \|\boldsymbol{p}_{t-1}^{\text{ball}} - \boldsymbol{p}_t^{\text{tar}}\|_2 - \|\boldsymbol{p}_t^{\text{ball}} - \boldsymbol{p}_t^{\text{tar}}\|_2$, clamped such that $0 < r_t^{\text{p}} < 1$, encourages the ball to get closer to the target. The contact reward $r_t^{\text{c}}$ encourages swinging the golf club to hit the ball, defined as:

$$r_t^{\text{c}} = \begin{cases} 1 \times \exp(-100 \times \|\boldsymbol{p}_t^{\text{ball}} - \boldsymbol{p}_t^{\text{club}}\|^2) & \text{if } C_{\text{cb}} = 0, \\ 1 & \text{if } C_{\text{cb}} = 1. \end{cases} \tag{2}$$

Here, $C_{\text{cb}} = 0$ indicates that the club has not made contact with the ball and $C_{\text{cb}} = 1$ indicates the club has made contact. The goal reward, $r_t^{\text{g}} = \exp(-0.1 \times \|\boldsymbol{p}_{t,xy}^{\text{ball}} - \boldsymbol{p}_{t,xy}^{\text{tar}}\|^2)$, encourages the ball to reach the target position in the x-y plane. In addition, we predict the ball's trajectory and provide a dense reward $r_t^{\text{pred}} = \exp(-0.1 \times \|\boldsymbol{p}^{\text{land}} - \boldsymbol{p}_{t,xy}^{\text{ball}}\|^2)$ based on the distance between the predicted landing point and the goal on the x-y plane [50]. The landing position, $\boldsymbol{p}^{\text{land}} = \left(x^{\text{land}}, y^{\text{land}}\right)$, can be calculated using the initial position and velocity as follows ($g$ is gravity):

$$x_{\text{land}} = x_0 + v_{0,x} \left( \frac{v_{0,z} + \sqrt{v_{0,z}^2 + 2gz_0}}{g} \right), \quad y_{\text{land}} = y_0 + v_{0,y} \left( \frac{v_{0,z} + \sqrt{v_{0,z}^2 + 2gz_0}}{g} \right) \tag{3}$$

Early termination is triggered if the ball moves backward, does not contact the golf club within 2 seconds, is too close to the humanoid's body, or the humanoid falls.

**High Jump**. The high jump environment simulates the Olympic high jump event, challenging the humanoid to clear a horizontal bar and land at a designated point. The setup mirrors official Olympic standards, with the bar positioned accordingly. Success is determined by clearing the bar cleanly (passing over the bar without touching it).

**Long Jump**. The long jump environment features a 20m runway followed by a jump area with the humanoid starting behind the jump line. The objective is to maximize the horizontal distance from the jump line to the point of landing, without crossing the jump line before takeoff. Success is defined as staying on track and jumping more than 1m.

**Hurdling**. The hurdling environment simulates the 110-meter hurdles with 10 hurdles, each 1.067 meters high. The first hurdle is 13.72 meters from the start, with 9.14 meters between each. The goal is to reach the finish line as fast as possible while clearing all hurdles.

**Javelin**. The javelin throw environment uses the humanoid model with articulated fingers (SMPL-X and Unitree G1). The humanoid's objective is to throw a javelin as far as possible. Success in this sport requires the humanoid to throw the javelin out of the starting area.

**Basketball (free-throw)**. Another task that leverages the articulated fingers (SMPL-X and Unitree G1) is the basketball free-throw environment. The humanoid begins with the ball initially positioned close to its hands. The objective is to successfully throw the basketball into the hoop, which is located

4.5 meters away and is 3 meters high. To succeed in this task the controller needs to successfully throw the ball into the hoop.

**Soccer (penalty kick)**. The player starts 13 meters from the goal, with the ball positioned 12 meters in front of the goal center. The objective is to kick the ball toward a randomly selected target within the goal. Success is achieved if the ball gets within 0.5 meters of the target.

**Tennis (single-player)**. In the single-player tennis environment, the humanoid practices returning shots on a standard-sized tennis court. The humanoid's right hand is replaced with an oval racket simulating a tennis racquet. Balls are launched from the opposite side of the court with randomized positions and trajectories, mimicking a variety of shots. The humanoid's objective is to successfully return these balls to randomly designated target areas on the opponent's side of the court. Success is measured by the ability to hit the ball towards the given target positions.

**Table Tennis (single-player)**. Similar to the single-player tennis environment, the table tennis task features a humanoid practicing return shots in a standard table tennis setup. The humanoid's right hand is replaced with a circular paddle instead of a tennis racket. As in tennis, balls are launched from the opposite side with varying parameters, but the smaller scale and faster pace of table tennis require quicker reflexes and more precise control. The objective remains the same: to return the balls to randomly designated target areas on the opponent's side of the table.

## 4.2 MULTI-PERSON SPORTS

In our competitive sports environments, we implement a basic adversarial self-play mechanism where two policies, initialized randomly, compete against each other to optimize their rewards. We use an alternating optimization strategy, inspired by [43], where we freeze one policy while training the other. This approach encourages the development of both offensive and defensive strategies in each policy. The effectiveness of this method is particularly evident in combat sports simulations such as boxing and fencing, as demonstrated in our `supplement`. Our evaluations on competitive sports focus on qualitative comparisons, showcasing behaviors each method learns.

**Soccer**. The soccer environment includes a ball, two goal posts, and the field boundaries. The field measures $32m \times 20m$. In this challenging task, two teams compete one against the other. We consider both 1v1 and 2v2 scenarios. The task observation $s_t^{\text{g-soccer}} \triangleq (p_t^{\text{ball}}, \dot{q}_t^{\text{ball}}, p_t^{\text{goal-post}}, p_t^{\text{ally-root}}, p_t^{\text{opp-root}})$ contains the root positions of the ally $p_t^{\text{ally-root}} \in \mathbb{R}^3$ (in 2v2) and opponents $p_t^{\text{opp-root}} \in \mathbb{R}^3$ (1 or 2), the position of the ball $p_t^{\text{ball}}$ and its velocity $\dot{q}_t^{\text{ball}}$, and position of the target goal post position $p_t^{\text{goal-post}}$.

To train the soccer policy, we propose the following dense reward components: $\mathcal{R}^{\text{soccer-match}}(s_t^{\text{p}}, s_t^{\text{g-soccer}}) \triangleq w^{\text{p2b}}r^{\text{p2b}} + w^{\text{b2g}}r^{\text{b2g}} + w^{\text{bv2g}}r^{\text{bv2g}} + w^{\text{point}}r^{\text{point}}$. (1) $r^{\text{p2b}}$: Encouraging the player to remain close to the ball. (2) $r^{\text{b2g}}$: $r^{\text{b2g}} = \|p_t^{\text{goal-target}} - p_{t-1}^{\text{ball}}\|_2 - \|p_t^{\text{goal-target}} - p_t^{\text{ball}}\|_2$ encourages the ball to move closer to the goal position. (3) $r^{\text{bv2g}}$: Encouraging an increase in ball velocity in the goal direction. To reward the controller for its contribution to the task, $r^{\text{b2g}}$ and $r^{\text{bv2g}}$ are zeroed out when the humanoid's distance to the ball is greater than 0.5m. (4) Finally, $r^{\text{point}}$: A one-time large bonus (or penalty) when a team scores a goal.

Notice that this is a rudimentary reward design compared to prior art [12] and serves as a starting point for further development.

**Tennis and Table Tennis (multi-player)**. The multi-player tennis and table tennis environment simulates a competitive 1v1 match on a tennis and table tennis court. We can either train models from scratch or initialize two identical single-player models as opponents, which can play back and forth using the shot-returning policy.

**Fencing**. For 1v1 fencing, each humanoid is equipped with a sword (replacing the right hand) and plays on a standard fencing field. The fencing reward encourages the agent to move toward the opponent and strike the pelvis, head, spine, chest, and torso of the opponent. The episode terminates if either of the humanoids falls or steps out of bounds.

**Boxing**. The objective of boxing is for each boxer to strike its opponent using fists while avoiding being hit. Valid striking areas include the head, torso, and upper body of the opponent.

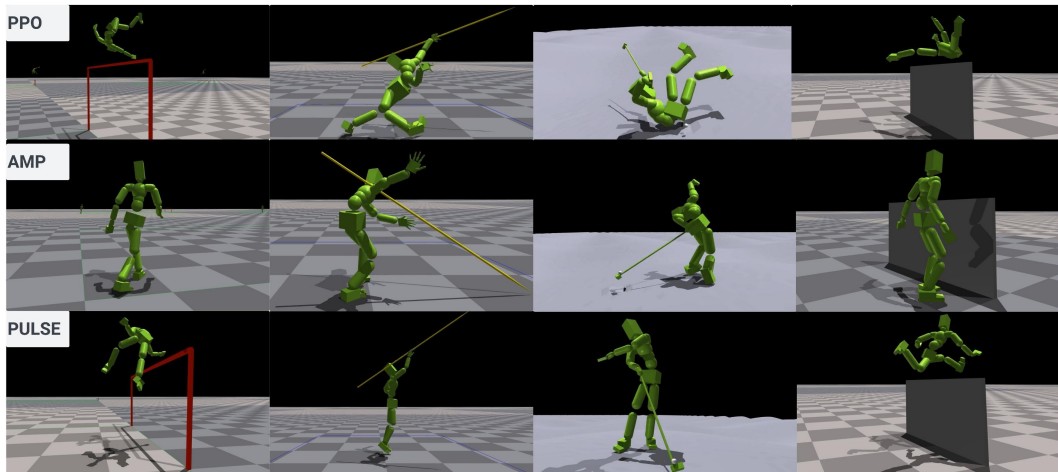

Figure 4: Qualitative results for high jump, javelin, golf, and hurdling. PPO and AMP try to solve the task using non-human-like behavior, while PULSE can discover human-like behavior.

For detailed state and reward definitions, please refer to Appendix B.1.

### 4.3 ACQUIRING HUMAN DEMONSTRATION

When existing MoCap dataset does not include motion for a particular sport, we propose a pipeline to acquire a small amount of human demonstration from videos. We utilize TRAM [42] for 3D motion reconstruction from Internet videos, providing robust global trajectory and pose estimation under dynamic camera movements, commonly found in sports broadcasting. Specifically, TRAM estimates SMPL parameters [13] which include global root translation, orientation, body poses, and shape parameters. We further apply PHC [14], a physics-based motion tracker, to imitate these estimated motions, ensuring physical plausibility. We find these corrected motions are significantly more effective as positive samples for adversarial learning compared to raw estimated results. More details and ablation are provided in the supplementary materials.

**Motion Retargeting**. To retarget motion from the SMPL format (both from MoCap or videos) to a humanoid robot (H1 and G1), we follow H2O [9] and perform a two-stage fitting process. First, we find a body shape $\beta'$ closest to the humanoid structure. We choose joints that have a correspondence between SMPL and humanoids and perform gradient descents on the shape parameter $\beta$ to minimize the joint distances using a common rest pose. After finding the optimal $\beta'$, given a sequence of motions expressed in SMPL parameters, we retarget motion from human to humanoid by minimizing the joint position differences using Adam optimizer [10].

## 5 EXPERIMENTS

**Implementation Details**. Simulation is conducted in Isaac Gym [19]. All task policies utilize three-layer MLPs with units [2048, 1024, 512]. All models can be trained on a single Nvidia RTX 3090 GPU in 1-3 days. The per-robot simulation details follow Section 3, and we provide additional details in about training (Appendix B.1) and hyperparamters (Appendix B.3).

**Algorithms**. We benchmark our simulated sports using state-of-the-art humanoid control methods. While not exhaustive, this selection provides a baseline for our challenging environments. We evaluate four key approaches: PPO, AMP, PULSE, and a combination of PULSE with AMP. Modifications to these methods (*e.g.* adding support to articulated fingers) can be found in Appendix B.4.

**PPO** [29] serves as our basic reinforcement learning algorithm and provides a baseline for comparison. We test two exemplary character animation frameworks that utilize human motion as prior. AMP **AMP** [24] utilizes a discriminator to provide an adversarial "style" reward using human demonstration as ground truth. During training, the discriminator provides a reward based on whether it thinks the state is "real" or "fake." It is jointly optimized with the policy using states generated from rolling out the policy and ground-truth states. **PULSE** [16] learns a reusable latent representation from a large-

Table 1: Evaluation on Long Jump, High Jump, Hurdling and Javelin. World records are in parentheses.

| | Long Jump (8.95m) | | High Jump (2.45m) | | | | Hurdling (12.8s) | | | Javelin (104.8m) | |
|---|---|---|---|---|---|---|---|---|---|---|---|
| Method | Suc Rate ↑ | Avg Dis ↑ | Suc Rate (1m) ↑ | Height (1m) ↑ | Suc Rate (1.5m) ↑ | Height (1.5m) ↑ | Suc Rate ↑ | Avg Dis ↑ | Time ↓ | Suc Rate ↑ | Avg Dis ↑ |
| PPO [29] | 53.6% | **19.42** | **100%** | **4.08** | **100%** | **4.11** | 57.6% | 108.9 | **11.22** | **100%** | **44.5** |
| AMP [24] | 0% | - | 0% | - | 0% | - | 0% | 13.24 | - | 0.31% | 2.03 |
| PULSE [16] | **100%** | 5.105 | **100%** | 2.01 | **100%** | 1.98 | **100%** | **122.1** | 17.76 | **100%** | 9.63 |

scale motion dataset via distilling from a universal humanoid motion imitator. PULSE uses a VAE-like latent space to model the conditional distribution of action based on proprioception. It has been shown to surpass previous methods [33, 25] in covering motor skills and applicability to downstream tasks. Unlike AMP, this reusable representation is employed via hierarchical RL to accelerate training while ensuring human-like behavior. In addition, PULSE can be effectively combined with AMP (denoted **PULSE with AMP**). *Both PULSE and AMP utilize human demonstration:* PULSE provides a way to reuse a wide range of learned behaviors, whereas AMP provides task-specific style reward.

We conduct all of our quantitative experiments on the SMPL family of humanoids as they exemplify our findings. Qualitative results for the SMPL humanoid and humanoid robots in the `supplement`.

**Metrics**. We provide quantitative evaluations for tasks with easily measurable metrics such as high jump, long jump, hurdling, javelin, golf, single-player tennis, table tennis, penalty kicks, and free throws. Qualitative assessments for tasks that are more challenging to quantify, such as boxing, fencing, and team soccer are provided in the `supplement`. Specifically, success rate (Suc Rate) determines whether an agent completes a sport according to set rules. Average distance (Avg Dis) indicates the extent an agent or object travels. For sports involving ball hits, such as tennis and table tennis, we record the average number of successful ball strikes (Avg Hits). Error distance (Error Dis) measures the distance between the intended target and the actual landing spot, applicable in sports like golf, tennis, and penalty kicks. Additionally, the hit rate in golf quantifies the success of striking the ball with the club. Evaluations are performed on 1000 trials.

### 5.1 BENCHMARKING POPULAR SIMULATED HUMANOID ALGORITHMS

In this section, we evaluate the performance of various control methods across our sports environments. We provide qualitative results in Fig. 4 and Fig. 5, and training curves in Fig. 6. To view qualitative results, including human-like soccer kick, boxing, high jump, *etc*., please see the `supplement`.

**Track & Field Sports (Without Video Data)**. We evaluate track and field sports, including long jump, high jump, hurdling, and javelin throwing, using a subset of the AMASS dataset for reference motions. For these sports, SOTA pose estimation methods fail to estimate coherent motion and global root trajectory from videos as players and cameras are both fast-moving. Thus, we utilize a subset of the AMASS dataset containing locomotion data [28] as reference motions for AMP. Since PULSE is pretrained on AMASS, we exclude PULSE with AMP from these tests.

As shown in Table 1, we observe consistent patterns in the performance across sports. AMP struggles to balance discriminator rewards with task completion, failing to execute the required tasks effectively. This is particularly evident in long jump and hurdling, where the agent either moves slowly or stops before completing the task. This failure occurs because the policy prioritizes discriminator rewards over task completion. If the task is too hard, the policy will use simple motion (such as standing still) to maximize the discriminator reward instead task reward.

PPO demonstrates the ability to achieve impressive physical feats, such as long distances in jumping or high clearances in hurdling. However, these actions are performed with unnatural, non-human-like motions. This highlights PPO's focus on task completion without regard for motion quality.

PULSE emerges as a balanced approach, consistently executing human-like motions across all events. Notably, in high jump, PULSE adopts a Fosbury flop technique without specific encouragement, likely leveraging skills it has encountered from the AMASS dataset, such as break-dancing. While PULSE successfully completes tasks with natural movements, it lacks the specialized skills required for top-tier performances in specific sports.

**Sports With Video Data**. We evaluate sports including golf, tennis, table tennis, and soccer penalty kicks using processed motion from videos as demonstrations for AMP and PULSE+AMP. Results are reported in Table 2 and Fig. 5.

Table 2: Evaluation on Golf, Tennis, Table Tennis, Penalty Kick and Free Throw

| | Tennis | | Table Tennis | | Golf | | Penalty Kick | | Free Throw |
| --- | --- | --- | --- | --- | --- | --- | --- | --- | --- |
| Method | Avg Hits ↑ | Error Dis ↓ | Avg Hits ↑ | Error Dis ↓ | Hit Rate ↑ | Error Dis ↓ | Suc Rate ↑ | Error Dis ↓ | Suc Rate ↑ |
| PPO [29] | 2.76 | **1.92** | 1.01 | **0.06** | 0% | - | 0.0% | - | **91.4%** |
| AMP [24] | **3.95** | 5.30 | 1.10 | 0.13 | **100%** | 1.43 | 0.0% | - | 0.0% |
| PULSE [16] | 2.48 | 3.50 | 0.74 | 0.19 | 99.9% | **1.29** | **76.6%** | **0.25** | 85.6% |
| PULSE [16] + AMP [24] | 2.62 | 3.64 | **1.83** | 0.23 | 99.9% | 2.18 | 27.5% | 0.27 | 89.8% |

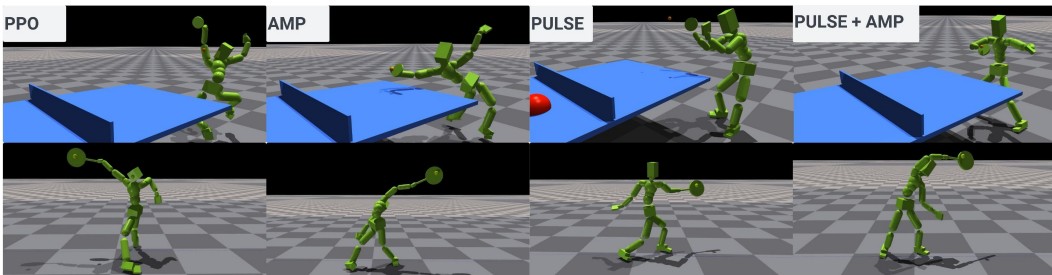

Figure 5: Qualitative results for table tennis and tennis. PPO and AMP result in inhuman behavior; PULSE can use human-like movement but PULSE + AMP result in behavior specific to the sport.

Similar to Track & Field, AMP struggles with balancing task and discriminator rewards even when it learns to complete the task. For instance, in tennis, it sacrifices motion quality at the moment of interaction (*e.g.* at the moment of ball contact) and reverts to natural movements when preparing for the next hit as shown in Fig. 5.

PPO consistently produces unnatural motions across all sports, despite sometimes achieving impressive metrics. Notably, in table tennis, PPO achieves very low error distances, as it lacks consistency and fails to return the second shots.

PULSE and PULSE+AMP show similar performance in most sports, with PULSE+AMP demonstrating particular effectiveness in table tennis. This is attributed to the combination of PULSE's pre-trained motor skills and the sport-specific guidance from video data, which is especially beneficial for sports requiring quick reactions.

For sports involving initiating contact with an object (golf, penalty kicks, free throws), PULSE and PULSE+AMP consistently outperform the other methods. This result aligns with findings in hierarchical reinforcement learning literature, which demonstrate that reusable skills facilitate more effective exploration [31, 32, 38]. In our case, PULSE's latent space serves as this hierarchical structure, enabling efficient navigation of the action space and overcoming the challenges of sparse rewards in these tasks.

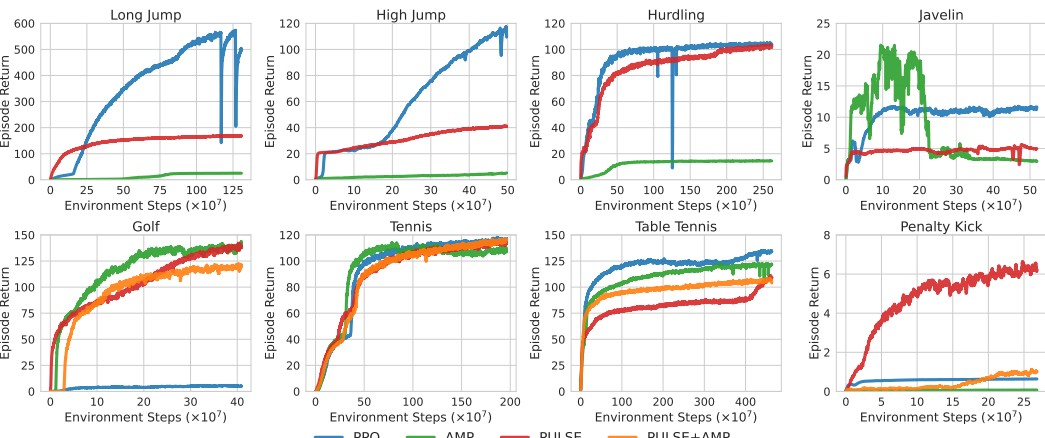

Figure 6: Learning curves on various tasks. Overall, for tasks that involve simple strategies (such as long jump and high jumps, PPO can solve the task more easily be leveraging non-human strategies. For tasks that require more coordinated movements, leveraging human demonstration using PULSE and AMP is more effective.

**Task Difficulty Diversity**. We also find that diversifying task difficulties is crucial, especially for challenging environments that struggle with early exploration. This strategy, while not a strict curriculum learning, involves exposing the model to a varied set of tasks with different challenge levels, sampled with equal probability throughout training. For instance, we randomly sample different hurdle heights in the hurdling task. Table 3 illustrates the impact of this task diversity in high jump and hurdling tasks using PULSE. Without task diversity, agents fail to jump over 1.5m and cannot complete the 110m hurdling.

When faced with challenging task settings, the policy can often be stuck in local minima and unable to learn rewards. Diversify the task difficulties provides a "ladder" that a policy can climb up such that it can learn to complete the harder tasks gradually by first solving the easier ones.

Table 3: Evaluation on diversifying task difficulties.

| Method | High Jump | | Hurdling | | |
| | Suc Rate (1m) | Suc Rate (1.5m) | Suc Rate | Avg Dis | Time |
| --- | --- | --- | --- | --- | --- |
| w/o | **100%** | 0% | 0% | 13.65 | - |
| w/ | **100%** | **100%** | **100%** | **122.1** | **17.76** |

Additional ablations on the quality of the human demonstration data and individual reward curves can be found in Appendix C.

## 6 LIMITATIONS, CONCLUSION, AND FUTURE WORK

**Limitations**. While HumanoidOlympics provides a large collection of simulated sports environments, it is far from being comprehensive. Certain sports are omitted due to simulation constraints (e.g., swimming, shooting, ice hockey, cycling) or their inherent complexity (e.g., 11-a-side soccer, equestrian events). Nevertheless, our framework is highly adaptable, allowing easy incorporation of additional sports like climbing, rugby, wrestling *etc*. Our initial design of rewards, though able to achieve sensible results, is also far from optimal. For competitive sports such as 2v2 soccer and basketball, our results also fall short of SOTA [12] which employs much more complex systems. In terms of supporting real-world humanoid embodiments, while we simulated our humanoid robots with realistic simulation parameters that have the potential to transfer to the real-world, we do not conduct any sim-to-real modifications (*e.g.* domain randomization) in designing our pipelines.

**Conclusion and Future Work**. We introduce HumanoidOlympics, a collection of sports environments for simulated humanoids. We provide carefully designed state and reward, and benchmark humanoid control algorithms and motion priors. We find that by combining expert reward design and powerful human motion prior, one can achieve human-like behavior for solving various challenging sports. Our humanoid's compatibility with the SMPL motion (either by design or through retargeting) also provides an easy way to obtain additional data from video for training, which we demonstrate to be helpful in training some sports. These well-defined simulation environments could also serve as valuable platforms for frontier models [17] to gain physical understanding. We believe that HumanoidOlympics provides a valuable starting point for the community to further explore physically simulated humanoids.

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

# Appendix

## A  INTRODUCTION

In the appendix, we provide comprehensive implementation details for HumanoidOlympics, including the reward designs for each sport environment, training procedures, and hyperparameters. Extensive qualitative results can be accessed on our supplement site, where we provide visualizations of all sports environments, humanoid embodiments, and training results based on our reward designs. Baseline results (PPO, AMP, PULSE, PULSE+AMP) are presented to support the quantitative findings discussed in the main paper. Furthermore, we offer visualizations of the reference motion extracted from in-the-wild videos. For our pipeline to acquire the human demonstration in the SMPL format, we conduct an ablation study evaluating the impact of employing a motion imitator (PHC [14]) as a refinement step. All code and trained models will be released. The submitted zip includes a lower-resolution version of the supplement site due to file size restrictions.

## B  IMPLEMENTATION DETAILS

### B.1  STATE, REWARDS, AND TERMINATION CONDITIONS

**High Jump**. For high jump, the humanoid's task is to leap over a horizontal bar positioned 20m ahead and 6m to the left of its starting point. The humanoid aims to reach the goal point $\boldsymbol{p}^{\text{g-high jump}} = (22, 6, 1)$ located 2 m behind the bar.

The high jump goal state $\boldsymbol{s}_t^{\text{g-high\_jump}} = (\boldsymbol{p}_t^b, \boldsymbol{p}_t^g)$ contains the positions of the bar $\boldsymbol{p}_t^b \in \mathbb{R}^3$ and the goal point $\boldsymbol{p}_t^g \in \mathbb{R}^3$.

The reward function is defined as follows:

$$\boldsymbol{\mathcal{R}}^{\text{high jump}}(\boldsymbol{s}_t^{\text{p}}, \boldsymbol{s}_t^{\text{g-high\_jump}}) \triangleq \begin{cases} 1 \times r_t^{\text{p}} & \text{if } \boldsymbol{p}_{t,x}^{\text{p}} \leq 19.5\text{m}, \\ 1 \times r_t^{\text{p}} + 1 \times r_t^{\text{h}} & \text{if } 19.5\text{m} < \boldsymbol{p}_{t,x}^{\text{p}} < 20.5\text{m}, \\ 1 \times r_t^{\text{p}} & \text{if } 20.5\text{m} \leq \boldsymbol{p}_{t,x}^{\text{p}}. \end{cases} \quad (4)$$

where $\boldsymbol{p}_{t,x}^{\text{p}}$ denotes the x-axis position. The height reward, $r_t^{\text{h}} = \boldsymbol{p}_{t,z}^{\text{p}}$, with $\boldsymbol{p}_{t,z}^{\text{p}}$ representing the z-axis position, incentivizes the humanoid to jump higher. The position reward, $r_t^{\text{p}} = \|\boldsymbol{p}_{t-1}^{\text{p}} - \boldsymbol{p}^{\text{g-high jump}}\|_2 - \|\boldsymbol{p}_t^{\text{p}} - \boldsymbol{p}^{\text{g-high jump}}\|_2$ (clamped to [0,1]), motivates the humanoid to reach the goal. An episode is terminated if the humanoid falls down, fails to leap over the bar, or moves beyond the designated run-up area.

**Long Jump**. In the long jump environment, the humanoid has a 20-meter runway before the jump line, which its feet should not exceed. The humanoid's goal is to reach the goal position, $\boldsymbol{p}^{\text{g-long jump}} = (30, 0, 1)$.

The goal state $\boldsymbol{s}_t^{\text{g-long\_jump}} \triangleq (\boldsymbol{p}_t^s, \boldsymbol{p}_t^l, \boldsymbol{p}_t^g)$ includes the position of the starting point $\boldsymbol{p}_t^s \in \mathbb{R}^3$, jump line $\boldsymbol{p}_t^l \in \mathbb{R}^3$, and the goal $\boldsymbol{p}_t^g \in \mathbb{R}^3$.

The training reward is defined as follows:

$$\boldsymbol{\mathcal{R}}^{\text{long jump}}(\boldsymbol{s}_t^{\text{p}}, \boldsymbol{s}_t^{\text{g-long\_jump}}) \triangleq \begin{cases} 1 \times r_t^{\text{p}} + 0.01 \times r_t^{\text{v}} & \text{if } \boldsymbol{p}_{t,x}^{\text{p}} \leq 20\text{m}, \\ 1 \times r_t^{\text{p}} + 0.01 \times r_t^{\text{v}} + 0.1 \times r^{\text{h}} + 30 \times r^{\text{l}} & \text{if } 20\text{m} < \boldsymbol{p}_{t,x}^{\text{p}}. \end{cases} \quad (5)$$

The position reward, $r_t^{\text{p}} = \|\boldsymbol{p}_{t-1}^{\text{p}} - \boldsymbol{p}^{\text{g-long jump}}\|_2 - \|\boldsymbol{p}_t^{\text{p}} - \boldsymbol{p}^{\text{g-long jump}}\|_2$ (clamped to [0,1]) encourages the humanoid to reach the goal point. The velocity reward, $r_t^{\text{v}} = \boldsymbol{v}_{t,x}^{\text{p}}$ prompts the humanoid to reach higher speed along the x-axis. The jump height reward $r_t^{\text{h}} = \boldsymbol{p}_{t,z}^{\text{p}}$ encourages the humanoid to jump higher after reaching the jump line. The jump length reward $r_t^{\text{l}} = \boldsymbol{p}_{t,x}^{\text{p}} - 20$ promotes longer final jump length. Each episode terminates if the humanoid falls or runs off the track.

**Hurdling**. In the hurdling task, the humanoid aims to reach a finish line 110m ahead while jumping over 10 hurdles, each 1.067m high. The first hurdle is placed 13.72m from the start, with subsequent hurdles spaced every 9.14m.

The goal state is defined as $\boldsymbol{s}_t^{\text{g-hurdling}} \triangleq (\boldsymbol{p}_t^h, \boldsymbol{p}_t^f)$, where $\boldsymbol{p}_t^h \in \mathbb{R}^{10 \times 3}$ and $\boldsymbol{p}_t^f \in \mathbb{R}^3$ includes the positions of these hurdles as well as the finish line.

The reward function is defined as $\boldsymbol{\mathcal{R}}^{\text{hurdling}}(\boldsymbol{s}_t^{\text{p}}, \boldsymbol{s}_t^{\text{g-hurdling}}) \triangleq r_t^{\text{distance}}$, which encourages the agent to run towards the finish line and clear each hurdle.

$$\boldsymbol{\mathcal{R}}^{\text{hurdling}}(\boldsymbol{s}_t^{\text{p}}, \boldsymbol{s}_t^{\text{g-hurdling}}) \triangleq 1 \times r_t^{\text{distance}} \quad (6)$$

The distance reward, $r_t^{\text{distance}} = \|\boldsymbol{p}_{t-1}^{\text{p}} - \boldsymbol{p}^{\text{g-hurdling}}\|_2 - \|\boldsymbol{p}_t^{\text{p}} - \boldsymbol{p}^{\text{g-hurdling}}\|_2$, is clamped to $[0, 1]$ and encourages the humanoid to get closer to the goal point. We terminate each episode if the character falls or runs off the track.

**Golf**. In the golf task, the humanoid is equipped with a golf club of dimensions of $0.05\text{m} \times 0.025\text{m} \times 0.02\text{m}$. The target location for the golf ball is positioned to the left of the humanoid, in the direction of the x-axis, at a distance ranging from 0m to 20m.

**Javelin**. For javelin throw, the humanoid is equipped with a javelin of length 2.7m.

The goal state is defined as $\boldsymbol{s}_t^{\text{g-javelin}} \triangleq (\boldsymbol{q}_t^{\text{obj}}, \boldsymbol{p}_t^r, \boldsymbol{p}_t^h)$, where $\boldsymbol{q}_t^{\text{obj}} \in \mathbb{R}^{13}$, includes the position, orientation, linear, and angular velocity of the javelin. $\boldsymbol{p}_t^r$ and $\boldsymbol{p}_t^h$ are the positions of the root and right hand.

Due to the complexity introduced by articulated fingers, the reward function $\boldsymbol{\mathcal{R}}^{\text{javelin}}$ is applied in three stages: first, the humanoid learns to hold the javelin stably; then, it learns to throw it; finally, the javelin flies as far as possible. A timer is used to differentiate the three stages. Specifically, $\boldsymbol{\mathcal{R}}^{\text{javelin}}$ is defined as follows:

$$\boldsymbol{\mathcal{R}}^{\text{javelin}}(\boldsymbol{s}_t^{\text{p}}, \boldsymbol{s}_t^{\text{g-javelin}}) \triangleq \begin{cases} 0.9 \times r_t^{\text{grab}} + 0.1 \times r_t^{\text{js}} & \text{if } t < 0.6s, \\ 0.9 \times r_t^{\text{goal}} + 0.05 \times r_t^{\text{s}} - 0.05 \times r_t^{\text{grab}} & \text{if } 0.6s \leq t < 1.2s, \\ 0.9 \times r_t^{\text{goal}} + 0.1 \times r_t^{\text{js}} & \text{if } 1.2s \leq t. \end{cases} \quad (7)$$

The reward for grasping $r_t^{\text{grab}} = \exp(-1 \times \|\boldsymbol{p}_t^{\text{right-hand}} - \boldsymbol{p}_t^{\text{javelin}}\|^2)$ encourages the hand to stay close to the javelin. The javelin stability reward $r_t^{\text{js}} = \exp(-1 \times \|\boldsymbol{q}_t^{\text{javelin}} - \boldsymbol{q}_t^{\text{javelin-default}}\|^2)$ encourages the 6 DoF pose of the javelin to remain close to the default pose, which faces forward and tilts 30 degrees upward, mimicking a flying pose. The humanoid stability reward, $r_t^{\text{s}} = \exp(-1 \times \|\boldsymbol{p}_t^{\text{root}}\|^2)$, encourages the humanoid to keep its root position fixed. The termination conditions vary according to the stage: during the grasping and throwing stages, the episode terminates if the javelin is too far from the right hand or deviates significantly from the default pose $\boldsymbol{q}_t^{\text{javelin-default}}$. During the flying stage, termination occurs if the javelin is too close to the right hand.

## B.2 MULTI-PERSON SPORTS

**Tennis**. For tennis, each humanoid is equipped with a circular racket with a 15cm radius, positioned 35cm away from the wrist, replacing the right hand. The court measures 23.77m in length and 8.23m

in width, mirroring the dimensions and layout of a real tennis court. The net height is 1m, and the simulated ball has a radius of 3.2cm. We design two tasks: a single-player ball return task, where the humanoid trains to hit balls launched randomly, and a 1v1 mode, where the humanoid competes against another humanoid. In the ball return task, the humanoid is positioned at the center of the baseline, with balls launched from the opposite side. The landing location is uniformly sampled on the opposite side and the ball launch velocity is randomly sampled.

The goal state is defined as $s_t^{\text{g-tennis}} \triangleq (p_t^{\text{ball}}, v_t^{\text{ball}}, p_t^{\text{racket}}, p_t^{\text{tar}}$, where $p_t^{\text{ball}} \in \mathbb{R}^3, v_t^{\text{ball}} \in \mathbb{R}^3, p_t^{\text{racket}} \in \mathbb{R}^3$ and $p_t^{\text{tar}} \in \mathbb{R}^3$, which includes the position and velocity of the ball, position of the racket and position of the target.

The reward function is defined as follows:

$$\mathcal{R}^{\text{tennis}}(s_t^{\text{p}}, s_t^{\text{g-tennis}}) \triangleq \begin{cases} 1 \times r_t^{\text{racket}} + 0 \times r_t^{\text{ball}}, & \text{if } C_{\text{rb}} = 0, \\ 0 \times r_t^{\text{racket}} + 1 \times r_t^{\text{ball}}, & \text{if } C_{\text{rb}} = 1. \end{cases} \tag{8}$$

Here, $C_{\text{rb}} = 0$ indicates that the racket has not made contact with the ball, and $C_{\text{rb}} = 1$ indicates the racket has made contact. $r_t^{\text{racket}} = \exp(-1 \times \|p_t^{\text{racket}} - p_t^{\text{ball}}\|^2)$ rewards the racket for getting closer to the ball. $r_t^{\text{ball}} = 1 + \exp(-1 \times \|p^{\text{land}} - p_t^{\text{tar}}\|^2)$ encourages the predicted landing location of the ball to be close to the target. Similar to the golf task, the landing location of the ball is calculated based on $p_t^{\text{ball}}$ and $v_t^{\text{ball}}$, providing a dense reward function to facilitate training [50]. Early termination occurs if the humanoid loses the point, either by failing to catch the ball or by hitting the ball out of bounds. In the 1v1 mode, two humanoids are placed on opposite sides of the court and the first ball is launched from the middle of the court, randomly directed at each player. The same reward function as the ball return task is used. To facilitate 1v1 training, the pre-trained model from the ball return task is used as a warm start. Similarly, the episode terminates if one player fails to catch the ball or returns the ball out of bounds.

**Table Tennis**. For table tennis, each humanoid is equipped with a circular paddle with an 8 cm radius, positioned 12 cm from the wrist, replacing the right hand. The table adheres to standard dimensions, featuring a playing surface 2.74 m in length and 1.525 m in width, standing 0.76 m high. The net is 15.25 cm high, and the table tennis ball has a radius of 2 cm. The setup includes a single-player ball return task and a 1v1 task.

The goal state is defined as $s_t^{\text{g-tennis}} \triangleq (p_t^{\text{ball}}, v_t^{\text{ball}}, p_t^{\text{racket}}, p_t^{\text{tar}})$, similar to tennis.

The reward function is designed similarly to tennis, except we define the ball reward as $r_t^{\text{ball}} = 1 + \exp(-1 \times \|p^{\text{land}} - p_t^{\text{tar}}\|^2) + N_{\text{hit}}$, where $N_{\text{hit}}$ counts the number of successful hits in one episode. This formulation is intended to encourage the humanoid to continuously hit the ball effectively. Unlike in golf and tennis, we calculate $p^{\text{land}}$ when it lands on the table at a height of 0.76 m. For early termination and the warm start in 1v1, we maintain the same setting as in the tennis task.

**Fencing**. For 1v1 fencing, similar to real-world fencing, the two players are confined to a 14m by 2m playground, where stepping out of the bound will reset the game.

The goal state is defined as $s_t^{\text{g-fencing}} \triangleq (p_t^{\text{opp}}, v_t^{\text{opp}}, p_t^{\text{sword}} - p_t^{\text{opp-target}}, \|c_t\|_2^2, \|c_t^{\text{opp}}\|_2^2, b)$, which contains the opponent's position body $p_t^{\text{opp}} \in \mathbb{R}^{24 \times 3}$, linear velocity $v_t^{\text{opp}} \in \mathbb{R}^{24 \times 3}$, the difference between target body position $p_t^{\text{opp-target}} \in \mathbb{R}^{5 \times 3}$ on the opponent and agent's sword tip position $p_t^{\text{sword}}$, normalized contract forces on the agent itself $\|c_t\|_2^2 \in \mathbb{R}^{24 \times 3}$ and its opponent $\|c_t^{\text{opp}}\|_2^2 \in \mathbb{R}^{24 \times 3}$, as well as the bounding box $b \in \mathbb{R}^4$.

The fencing reward is structured similarly to the boxing setup in NCP [52]:

$$\mathcal{R}^{\text{fencing}}(s_t^{\text{p}}, s_t^{\text{g-fencing}}) \triangleq 0.1 \times r_t^{\text{facing}} + 0.1 \times r_t^{\text{vel}} + 0.6 \times r_t^{\text{strike}} + 1 \times r_t^{\text{point}}. \tag{9}$$

The facing reward $r_t^{\text{facing}}$ penalizes deviation from facing the opponent's root position $p_t^{\text{opp-root}}$. The velocity reward, $r_t^{\text{vel}}$, encourages the x-y plane linear velocity to be directed towards the opponent's root position $p_t^{\text{opp-root}}$. The strike reward, $r_t^{\text{strike}} = \exp(-10 \times \arg\min \|p_t^{\text{sword}} - p_t^{\text{opp-target}}\|^2)$, encourages the swordtip to get closer to the target body parts $p_t^{\text{opp-target}}$, which include the pelvis, head, spine, chest, and torso. If there is contact with the target body part with sufficient force, a positive reward is provided:

$$r_t^{\text{point}} = \begin{cases} 1 & \text{if } \arg\min \|\boldsymbol{p}_t^{\text{sword}} - \boldsymbol{p}_t^{\text{opp-target}}\|^2 \leq 0.1 \text{ and contact force} \geq 50\text{Nm}, \\ 0 & \text{otherwise.} \end{cases} \tag{10}$$

Our fencing agents are trained using competitive self-play, as introduced in the main paper.

**Boxing**. For boxing, the humanoid competes in a boxing ring measuring 5m by 5m. The humanoid's right hand is replaced with a sphere of 8cm radius.

The goal state is similar to fencing: $\boldsymbol{s}_t^{\text{g-boxing}} \triangleq (\boldsymbol{p}_t^{\text{opp}}, \boldsymbol{v}_t^{\text{opp}}, \boldsymbol{p}_t^{\text{hand}} - \boldsymbol{p}_t^{\text{opp-target}}, \|\boldsymbol{c}_t\|_2^2, \|\boldsymbol{c}_t^{\text{opp}}\|_2^2)$ but without the bounding box information.

The boxing reward function has the same composition as fencing, except that the sword tip position $\boldsymbol{p}_t^{\text{sword}}$ is replaced by the hand position $\boldsymbol{p}_t^{\text{hand}}$. Our boxing agents are also trained using competitive self-play.

**Soccer**. The soccer field measures 32m in length and 20m in width. Each goal is 4m wide and 2m tall. The ball has a diameter of 11.5 cm and weighs 450 grams.

For the penalty kick task, the reward function $\mathcal{R}^{\text{soccer-kick}}(\boldsymbol{s}_t^{\text{p}}, \boldsymbol{s}_t^{\text{g-kick}}) \triangleq w^{\text{p2b}}r^{\text{p2b}} + w^{\text{b2g}}r^{\text{b2g}} + w^{\text{bv2g}}r^{\text{bv2g}} + w^{\text{b2t}}r^{\text{b2t}} - c_t^{\text{no-dribble}}$ is divided into stages based on whether the ball is moving toward the goal. Specifically, we define a "closer to goal" variable as $g_t^{\text{ball-to-goal}} = \|\boldsymbol{p}_t^{\text{goal-target}} - \boldsymbol{p}_{t-1}^{\text{ball}}\|_2 - \|\boldsymbol{p}_t^{\text{goal-target}} - \boldsymbol{p}_t^{\text{ball}}\|_2$, which indicates whether the ball is getting closer to the goal. The full reward function is defined as follows:

$$\mathcal{R}^{\text{soccer-kick}}(\boldsymbol{s}_t^{\text{p}}, \boldsymbol{s}_t^{\text{g-kick}}) \triangleq \begin{cases} 0.4 \times r^{\text{p2b}} - c_t^{\text{no-dribble}} & \text{if } g_t^{\text{ball-to-goal}} \leq 0, \\ 0.1 \times r^{\text{b2g}} + 0.1 \times r^{\text{bv2g}} + 0.8 \times r^{\text{b2t}} - c_t^{\text{no-dribble}} & \text{otherwise.} \end{cases} \tag{11}$$

Essentially, if the ball is not moving toward the goal, the humanoid is encouraged to move toward the ball; if the ball is moving, the agent is rewarded for shooting the ball toward the target in the goal post. The player-to-ball reward, $r^{\text{p2b}} = \|\boldsymbol{p}_{t-1}^{\text{root}} - \boldsymbol{p}_{t-1}^{\text{ball}}\|_2 - \|\boldsymbol{p}_t^{\text{root}} - \boldsymbol{p}_t^{\text{ball}}\|_2$, is a point-goal reward [44]. The ball-to-goal reward $r^{\text{b2g}} = \|\boldsymbol{p}_t^{\text{goal-target}} - \boldsymbol{p}_{t-1}^{\text{ball}}\|_2 - \|\boldsymbol{p}_t^{\text{goal-target}} - \boldsymbol{p}_t^{\text{ball}}\|_2$ encourages the ball to move closer to the goal position. The ball-velocity-to-goal reward $r^{\text{bv2g}}$ incentivizes the ball velocity toward the goal position. The ball-to-target reward $r^{\text{b2t}}$ predicts the landing position of the ball in the net based on its current velocity and position, providing a reward if the ball is close to the target. Finally, $c_t^{\text{no-dribble}}$ penalizes the humanoid if its root position is over the ball's spawning point.

The team play (1v1 and 2v2) soccer tasks use similar rewards as the penalty kick task. The reward function for team play is $\mathcal{R}^{\text{soccer-match}}(\boldsymbol{s}_t^{\text{p}}, \boldsymbol{s}_t^{\text{g-soccer}}) \triangleq w^{\text{p2b}}r^{\text{p2b}} + w^{\text{b2g}}r^{\text{b2g}} + w^{\text{bv2g}}r^{\text{bv2g}} + w^{\text{point}}r^{\text{point}}$, where $r^{\text{p2b}}$, $r^{\text{b2g}}$ are the same as in the penalty kick. $r^{\text{point}}$ provides a one-time bonus for scoring.

**Basketball**. The basketball environment is similar to soccer except that it utilizes the SMPL-X humanoid with articulated fingers. In the free-throwing task, the ball is initialized between the humanoid's hands.

The goal state for this task is defined similarly to that of the soccer penalty kicks.

The free throw reward is defined as: $\mathcal{R}^{\text{free-throw}}(\boldsymbol{s}_t^{\text{p}}, \boldsymbol{s}_t^{\text{g-soccer}}) \triangleq 0.5 \times r^{\text{ballvel}} + 0.5 \times r^{\text{bv2g}} + r^{\text{basket}}$. The basketball velocity reward $r^{\text{ballvel}} = \exp(-0.1 \times \|\boldsymbol{v}_t^{\text{ball}} - \boldsymbol{v}_t^{\text{ball-desired}}\|_2^2)$ encourages the ball's velocity to be close to the desired velocity to reach the goal. The desired velocity, $\boldsymbol{v}_t^{\text{ball-desired}}$, is computed using the goal position $\boldsymbol{p}_t^{\text{goal-target}}$, and the ball position $\boldsymbol{p}_t^{\text{ball}}$, with the following physics equations:

$$T_t^{\text{reach}} = \sqrt{\frac{2 \times \|(\boldsymbol{p}_t^{\text{ball}} - \boldsymbol{p}_t^{\text{goal-target}})_z\|_2}{g}} \,, \quad \boldsymbol{v}_{t,xy}^{\text{ball-desired}} = \frac{\|(\boldsymbol{p}_t^{\text{ball}} - \boldsymbol{p}_t^{\text{goal-target}})_{xy}\|_2}{T_t^{\text{reach}}}$$

$$\boldsymbol{v}_{t,z}^{\text{ball-desired}} = \frac{(\boldsymbol{p}_t^{\text{ball}} - \boldsymbol{p}_t^{\text{goal-target}})_z + 0.5 \times g \times (T_t^{\text{reach}})^2}{T^{\text{reach}}}. \tag{12}$$

The ball-velocity-to-goal reward $r^{\text{bv2g}}$ encourages the velocity to be directed towards the goal position. The basket reward, $r^{\text{basket}}$, provides a one-time reward if the ball passes through the basket.

Table 4: Hyperparameters for training each baseline used in HumanoidOlympics. We use the same set of hyperparamters for *each sport*. Notice that AMP and PULSE uses PPO as the optimization method but add respective motion priors (as reward or motion representation). $\sigma$: fixed variance for policy. $\gamma$: discount factor. $\epsilon$: clip range for PPO. $w_{\text{disc}}$ and $w_{\text{task}}$: weights for discriminator and task rewards.

| | Batch Size | Learning Rate | $\sigma$ | $\gamma$ | $\epsilon$ | MLP-size | $w_{\text{disc}}$ | $w_{\text{task}}$ | # of samples |
|---|---|---|---|---|---|---|---|---|---|
| PPO [29] | 1024 | $5 \times 10^{-4}$ | 0.05 | 0.99 | 0.2 | [2048, 1024, 512] | 0 | 1 | $\sim 10^9$ |
| AMP [24] | 1024 | $5 \times 10^{-4}$ | 0.05 | 0.99 | 0.2 | [2048, 1024, 512] | 0.5 | 0.5 | $\sim 10^9$ |
| PULSE [16] | 1024 | $5 \times 10^{-4}$ | 0.3 | 0.99 | 0.2 | [2048, 1024, 512] | 0 | 1 | $\sim 10^9$ |
| PULSE [16] + AMP [24] | 1024 | $5 \times 10^{-4}$ | 0.3 | 0.99 | 0.2 | [2048, 1024, 512] | 0.5 | 0.5 | $\sim 10^9$ |

Team-play basketball has a similar reward design as soccer. The team-play basketball task is highly challenging due to the difficulty of picking the ball up, which is more complex than kicking a ball. Thus, while we support 1v1 and 2v2 team-play basketball, our preliminary reward design does not yield interesting behavior, unlike in soccer.

### B.3    HYPERPARAMTERS

Training hyperparameters are provided in Table 4. We use the same set of hyperparameters to train *all* of our sports environments, highlighting the advantage of employing a unified humanoid embodiment for simulated sports.

### B.4    DETAILS ABOUT ALGORITHMS

For PULSE [16] and AMP [24], we use the official implementations. For PULSE [16], we employ the publicly released model without modification, which is pre-trained on the AMASS dataset. We follow a similar setup for downstream tasks in PULSE, using the frozen prior $\mathcal{P}_{\text{PULSE}}$, decoder $\mathcal{D}_{\text{PULSE}}$, and residual action representation. Since PULSE only includes trained models for the SMPL-based models, we train SMPL-X humanoid-based models following the official implementation. Specifically, we train a humanoid motion imitator following PHC [14], and distill motor skills into a 48-dimensional latent space [15] (instead of 32-D, to accommodate articulated fingers). PULSE provides an action space for hierarchical RL and can be integrated with AMP. For PULSE+AMP, the AMP reward offers additional style guidance for the humanoid, which is particularly beneficial for tasks such as table tennis. However, we find that the demonstration sequences used for AMP need to be task-specific (*e.g.* contains only a swinging motion); otherwise, the discriminator reward can overpower the task reward and lead to undesired behavior (as seen in the free kick results).

## C    ADDITIONAL ABLATIONS

We conducted an ablation study to evaluate the role of physics-based tracking (w/ PHC) in acquiring human reference motion. Specifically, we used the pose estimation results directly from TRAM [42] as positive samples for the discriminator during policy training (w/ PHC). Our experiments were performed in the context of table tennis. As shown in Table 5, we found that providing video data without PHC leads to significantly lower performance compared to using PHC, similar to the results obtained using only PULSE.

We observe that when the quality of the provided reference motion is poor (e.g., with significant noise in position, and drastic velocity changes), the model struggles to effectively utilize the reference motion as style guidance to achieve natural movements. In contrast, employing physics-based tracking to refine pose estimates from in-the-wild videos results in physically plausible motion, which significantly aids in policy learning.

Table 5: Ablation study on PHC.

| Method | Table Tennis | |
|---|---|---|
| | Avg Hits ↑ | Error Dis ↓ |
| PULSE | 0.74 | 0.19 |
| PULSE+AMP, w/o PHC | 0.91 | **0.18** |
| PULSE+AMP, w/ PHC | **1.83** | 0.23 |

We conduct another ablation study on reward design, specifically presenting the learning curves of different reward terms in tennis learning in Fig. 7. To better visualize the differences, we apply a logarithmic transformation to the rewards. Initially, the paddle reward increases rapidly, but once the

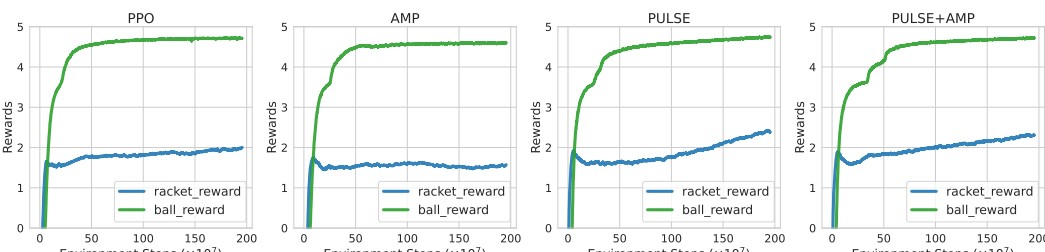

Figure 7: Learning curve on the individual reward terms for tennis.

racket consistently hits the ball, the ball reward begins to rise quickly. We observe that for AMP, the ball reward is slightly lower, reflecting lower accuracy in hitting the ball (larger Error Dis in Table 2), and its smaller paddle reward is observed because, through visualization, we find that the agent tends to swing the racket faster, thus quickly entering the contact phase. This could be why AMP manages to hit more balls continuously.

## D   BROADER SOCIAL IMPACT

We propose HumanoidOlympics, a collection of sports environments for simulated humanoids. These environments can be used to benchmark learning algorithms, discover new humanoid behaviors, create animations, and more. The potential negative social impact includes the risk of generating animations that could be used to create DeepFakes. Positive social impact includes the development of intelligent and collaborative agents, advancements in robot learning, discovery of new sports techniques, and the generation of immersive and physically realistic animations.

