# OpenReview forum: "HumanoidOlympics: Sports Environments for Physically Simulated Humanoids"
_ICLR.cc/2025/Conference — Submitted to ICLR 2025_

### Official Review · Reviewer_ayc8 · 2024-10-21

**Soundness:** 3
**Presentation:** 3
**Contribution:** 3
**Rating:** 6
**Confidence:** 4

**Summary:**

This paper presents reinforcement learning environments for sports by humanoid robot agents. These environments contain some humanoid robot models and designed states and rewards for learning sports. The paper provides the benchmark using a state-of-the-art humanoid control learning method. Learning in the environment uses human demonstrations.

**Strengths:**

- Reinforcement learning environments for sports by humanoid robot agents are useful for evaluating learning algorithms, understanding human sports, and helping human sports.
- The designed states, designed rewards, and benchmarks enable researchers to use the environments and evaluate their methods.
- The environments have task difficulty diversity. Thus, the researchers evaluate the performance of their method using the environments.

**Weaknesses:**

The design of the proposed environments looks straightforward. Highlighting the difficulty of designing the rewards and states could help readers understand the paper's contribution more. Do agents using states and rewards other than the proposed ones fail to solve tasks? Do the environments equipped with states and rewards other than the proposed ones fail to represent real sports?

**Questions:**

- How is it challenging to design states and rewards and select sports for the environments?
- Did the authors face particular challenges in designing states and rewards for any sports?
- What are the criteria for selecting which sports to include?
- Which unique difficulty did the authors encounter when modeling certain sports compared to others?
- How did the authors balance realism with computational feasibility?

- Do the environments have specific advantages that other environments rarely have?
- How does HumanoidOlympics differ from or improve upon specific existing humanoid or sports simulation environments?
- Are there particular aspects of human-like motion or sports-specific challenges that these environments capture better than others?

---

> ### Author Response · Authors · 2024-11-25
> **Author Response to Reviewer ayc8**
>
> Thank you for your constructive and positive review! Here, we address your concerns and hope you can raise the score.
>
> **Do agents using states and rewards other than the proposed ones fail to solve tasks?**
>
> We do not claim that our reward and state design are optimal; we extensively search for the recipe that leads to human-like behavior and high success rates. Without expert reward designs, the tasks are not successfully solved.
>
> **Do the environments equipped with states and rewards other than the proposed ones fail to represent real sports?**
>
> We do not claim that our reward and state designs are the optimal or only solution to represent sports. We open-source our code so that the community can build upon it to achieve better results.
>
> **How is it challenging to design states and rewards?**
>
> To create human-like behavior for simulated humanoids, extensive engineering is needed to design rewards and states to guide the learning of the behaviors. Some sports (ping-pong, tennis, fencing, etc.) have standalone full papers designing specialized rewards [1, 2, 3]. We achieve comparable visual quality while unifying 10+ sports under the same embodiment and training pipeline.
>
> **Did the authors face particular challenges in designing states and rewards for any sports?**
>
> One example of the difficulty of designing the state/rewards is the penalty kick. While a simple task in concept, how to encourage the humanoid to "kick" is challenging. A simple reward like “ball reaching the goal” would be too sparse. Rewards encouraging the humanoid to get closer to the ball would lead to the humanoid running with the ball instead of kicking. A simple “ball reaching the goal” is also too sparse for reaching the goal target, as the ball needs to fly at certain angles and speed to reach the target. Our design balances all these factors and calculates the ball's trajectory based on the current velocity to achieve the kicking motion. Notice that many of our results (Fosbury high jump, javelin throw, golfing, etc.) are novel tasks and results.
>
> **What are the criteria for selecting which sports to include?**
>
> We divide sports based on competitive multi-person ones and single-person ones. We pick competitive ones that either involve an object (ping pong, tennis) or player-to-player actions (boxing and fencing). For single-person ones, we implement popular track-and-field sports.
>
> **Which unique difficulty did the authors encounter when modeling certain sports compared to others?**
>
> See above for an example. Compared to other efforts in simulated sports, we design environments that share the same embodiment (e.g. SMPL, H1, G1). and sometimes motion prior (e.g., PULSE). This leads to the need to consider the task definition and compatible training pipelines for all sports.
>
> **How did the authors balance realism with computational feasibility?**
>
> We use the same simulation parameters as prior art for character animation [2]. For real humanoid robots, we use the same parameters as prior sim-to-real efforts [4].
>
> **Do the environments have specific advantages that other environments rarely have?**
>
> We present many tasks not presented in the literature before (long jump, high jump, javelin throw, golfing, etc.). For the ones that have been presented, we unify the embodiment such that the same humanoid can be used to solve all of these tasks. This makes obtaining demonstrations easier.
>
> **How does HumanoidOlympics differ from or improve upon specific existing humanoid or sports simulation environments?**
>
> There are very few existing sports simulation environments. Prior efforts in simulated humanoid sports are isolated and specialized, while ours uses common humanoids to learn all sports. We also focus on leveraging human motion priors and demonstrations.
>
> **Are there particular aspects of human-like motion or sports-specific challenges that these environments capture better than others?**
>
> We believe that the HumaonidOlympic Games offer a collection of sports and shared embodiments that have never been demonstrated before in the community.  Current RL benchmarks [5, 6, 7] often focus on locomotion tasks such as moving forward and traversing terrain. We offer much more diverse and close-to-real-life tasks.
>
>
> [1] Wang et al. "Strategy and skill learning for physics-based table tennis animation." SIGGRAPH 2024.
>
> [2] Zhang et al. "Learning physically simulated tennis skills from broadcast videos." SIGGRAPH 2023.
>
> [3] Won et al. "Control strategies for physically simulated characters performing two-player competitive sports." TOG 2021
>
> [4] He, Tairan, et al. "OmniH2O: Universal and Dexterous Human-to-Humanoid Whole-Body Teleoperation and Learning." CORL 2024
>
> [5] Brockman, G. "OpenAI Gym."
>
> [6] Makoviychuk, Viktor, et al. "Isaac gym: High performance gpu-based physics simulation for robot learning."
>
> [7] Tunyasuvunakool, Saran, et al. "dm_control: Software and tasks for continuous control." Software Impacts 2020.

---

### Official Review · Reviewer_QxJF · 2024-11-03

**Soundness:** 3
**Presentation:** 3
**Contribution:** 2
**Rating:** 5
**Confidence:** 4

**Summary:**

This paper proposes a simulation environment for the simulation and learning of various sports tasks using several humanoid robot designs. Through dense reward design and learning from demonstration schemes, which use two different methods from the literature called PULSE and AMP, the authors show that the RL algorithm PPO can learn more human-like motion for these robots. The humanoids are compatible with SMPL motion, and the SMPL human motion parameterization provides (together with the use of algorithms for the transfer task) a relatively easy way to transfer motion from humans to the humanoids. Various comparisons are made in the Experiments section using PPO, PULSE, AMP and their combinations, showing the effects/contributions coming from the different algorithmic components.

===== POST-REBUTTAL RESPONSE =====

I thank the authors for their rebuttal and for answering my questions. I am not going to raise the score, because after careful reading of other reviews and the general rebuttal, I tend to think that the contribution is a bit marginal. I would have liked to see more effort on the reward design / simulation side to address some of my concerns.

However I liked the writing style and the presentation of the material, I encourage the authors to continue working on it [if not accepted] and either (i) submit to a non-learning-focused conference as is (or with small modifications) or (ii) expand on the learning contributions (e.g. more RL focus, sim-to-real focus etc.) and submit to a ML conference.

**Strengths:**

There seems to be a clear, albeit modest, contribution to the literature: having a unified simulation environment for the testing/simulation of various sports tasks for various humanoids is going to be useful for the community, as we try to push current RL methods to higher dimensional systems and more complex scenarios. Moreover, the paper is well written and presents their work clearly.

**Weaknesses:**

However there are some limitations and weaknesses, some of which could be addressed / improved perhaps in the rebuttals:

- The methods are not introduced adequately, in particular it is not clear what PULSE and AMP are doing. The appendix discusses how the authors used PULSE for their work but it is not enough to form a clear picture of what PULSE is doing, and AMP is not discussed. There needs to be a longer, more self-contained discussion I think, some of which should be streamlined with the main text.
- Only PPO is used as the baseline RL algorithm in the Experiments but the paper in the introduction and conclusion states that they 'benchmark' the humanoids control algorithms, which is clearly an overstatement. If the authors claim that benchmarking is a contribution of the paper, then we need more careful investigation of different RL (or non-RL) approaches.
- There should be ablations and more experimentation of reward design, especially the effect (of different or varying reward designs) on the human-likeness of the learned motions and the effect on the task success rates should be discussed. See one of the questions below for more details on this point.
- As a more minor point, the experiments use only a three-layer MLP to learn the RL policy, which may be too limiting.

The last three points could be (at least partially) addressed by a much more careful and detailed ablations (some of which needs to go to / streamlined with the main text). There could be extensive ablations of different models, different state / reward representations, and most importantly effects of different RL algorithms. As of now, what we can learn from the paper is quite limited, except to know that we can simulate many different tasks with humanoids (which is also a contribution, but not the only or most important one, according to the text).

**Questions:**

Some questions and minor points/corrections:
- "Embodiement" -> Embodiment
- "Another important challenge of working with simulated humanoids is the ease of obtaining human demonstrations." Is it a challenge, or is it an opportunity not sufficiently exploited (till now)?
- Would be nice to mention why SMPL humanoids have 3DOF actuation per joint.
- "To overcome this, we design dense reward functions to guide the learning process." But wouldn't this be interfering with the original task? Can you toggle them off if needed? It's not necessary to overly-constrain the robots: we might miss out on some interesting strategies if we over-specify the reward.
- A contrary point to the above perhaps with respect to the dense rewards: instead of learning from demonstrations (or in addition to), can you not add dense rewards to encourage more human-like or realistic movement? For instance, you can penalize jerky motion, penalize joint limit violations or apply constrained RL, etc.
- "We modify the humanoid model by replacing its right hand with a 1.4-meter-long golf club." Why not also learn the holding mention, does it make the task too difficult? (similarly for tennis and table tennis, I think there could be an option to add the humanoid hand back)
- p_t^c or c_t^b for the club position?
- terrain height o_t is not changing w.r.t time I guess?
- I suggest removing 1 x notation in eq. (1) and elsewhere for improved readability. (especially so given that you don't discuss scaling)
- is eq. (1) dense meaning that there's a reward generated at each t? (so every 1/30 seconds for a robot operated at 30Hz)
- you don't assume any air friction in eq. (3) it seems.
- check line 304, (1) could come right after "where".
- notation for soccer could be simplified.
- "inhuman behavior" in Figure 4: inhuman has a negative connotation, use e.g. less/not human-like
- what are 'qualitative results' ?
- is PPO not using human demonstrations? That wasn't clear to me from the text initially. Likewise not clear if other methods use PPO as the RL algorithm in the main text (appendix mentions it only).
- "All task policies utilize three-layer MLPs with units [2048, 1024, 512]." How did you decide on this parameterization? What are the inputs to the networks? The observations of the states?
- why only use PPO/AMP/PULSE out of many options? That wasn't clear.
- In general we're lacking critical information about these algorithms to follow the text closely. For instance we have to go to the appendix to: "Notice that AMP and PULSE uses PPO as the optimization method but add respective motion priors (as reward or motion representation)." The main text should be as stand alone / independent as possible from the appendix (I realize it is hard in ML conferences but we should try).
- "PULSE adopts a Fosbury flop technique without specific encouragement" adopts or learns / behaviour emerges?
- Figure 5: "PPO and AMP result in inhuman behavior" What are these "unnatural non-human-like motions" (line 426)? Can you not engineer the rewards easily so that human-like behaviour emerges (or likelihood is increased)?
- "diversifying diverse task difficulties" -> choose one diverse
- Table 3 is not clear, what is the difference between "w/o" vs. "w/"?
- "We find that by combining expert reward design and powerful human motion prior, one can achieve human-like behavior for solving various challenging sports" If this is one of the (and perhaps the strongest, according to the text) contributions of the paper, we'd expect it to be more carefully presented: e.g. by comparing against more RL methods / more investigations on reward design. Moreover, what is the use-case of achieving more human-like behavior? That is not discussed.
- To continue the point above, one can clearly see for instance that PPO generates ridiculous looking movements, that could never be tried on any hardware. That is because some limits of the robot (jerk limits, or joint limits perhaps) would be exceeded immediately, which can be quite 'dangerous'. So we see that safety can be an important factor, but safety can be achieved through other means than including motion priors, e.g., using constraints, or better reward design. In fact safety may not be achieved merely through human motion priors.
- task observation (line 160) and goal state seem to be used interchangably in the appendix, making it quite confusing to understand what is the goal of the particular task. For instance in tennis (line 817) only p^{tar} is the goal state whereas the whole task observation is presented as the goal state.

---

> ### Author Response · Authors · 2024-11-25
> **Author Response to Reviewer QxJF (1/3)**
>
> Thank you for your constructive feedback!  Here we address your concerns in the hope that the reviewer raises the score.
>
> **Purpose of AMP and PULSE**
>
> Thank you for the suggestion! We have added an additional discussion of the algorithm we deployed in the Experiments section. We include it here as well:
>
> > We test two exemplary character animation frameworks that utilize human motion as prior. AMP utilizes a discriminator to provide an adversarial "style" reward using human demonstration as ground truth. During training, the discriminator provides a reward based on whether it thinks the state is “real” or “fake.” It is jointly optimized with the policy using states generated from rolling out the policy and ground-truth states. PULSE  learns a reusable latent representation from a large-scale motion dataset via distilling from a universal humanoid motion imitator. PULSE uses a VAE-like latent space to model the conditional distribution of action based on proprioception. It has been shown to surpass previous methods in covering motor skills and applicability to downstream tasks. Unlike AMP, this reusable representation is employed via hierarchical RL to accelerate training while ensuring human-like behavior.
>
> **Baseline RL Methods**
>
> Excellent suggestions! We want to clarify that we are benchmarking popular character animation methods such as AMP and PULSE, which utilize data-driven priors to achieve human-like motion. Other RL approaches (either model-free or model-based), such as SAC, TD-MPC, Dreamer, etc., do not provide motion priors and lead to jarring humanoid behaviors [1]. Most character animation and sim-to-real approaches use PPO due to its simplicity and scalability. In this work, we focus on benchmarking data-driven approaches that utilize human motion as prior, which leads to human-like behaviors.
>
> **More Experimentation of Reward Designs **
>
> We provide a detailed description of each reward component and intuition in Appendix B. We believe the environments are meant to serve as a starting point for the community to develop more human-like behavior for simulated humanoids. Our criteria for choosing the reward is completing the task, and the reward designs are by no means the most optimal one. We feel that analyzing each set of manually designed rewards for the 10+ tasks we provide would detract from our main message: sports environments are challenging, and leveraging human motion as prior is important.
>
> **Three Layer MLP**
>
> Three-layer MLP for RL is widely used for character animation and sim-to-real robotics [2, 3, 4, 5] due to the high inference speed requirement for RL training. We can certainly explore bigger networks; however, network size is not the bottleneck for RL training.

---

> ### Author Response · Authors · 2024-11-25
> **Author Response to Reviewer QxJF (2/3)**
>
> **Questions:**
>
> > Q: Is it a challenge, or is it an opportunity not sufficiently exploited (till now)?
>
> A: It is not sufficiently exploited since obtaining high-quality human demonstration data is challenging. High-quality demonstration needs to be in the form of a global 3D human pose [6], which is notoriously hard to estimate accurately due to camera motion and depth ambiguity. Retargeting them to humanoids is also an active research area [7].
>
> > Q: Would be nice to mention why SMPL humanoids have 3DOF actuation per joint.
>
> A: The SMPL body model uses a 3DOF joint for each one of its joints. To make our humanoid compatible with the data format, we design our SMPL humanoid to have 3DoF per joint.
>
> > Q: But wouldn't this be interfering with the original task? Can you toggle them off if needed? It's not necessary to overly-constrain the robots: we might miss out on some interesting strategies if we over-specify the reward.
>
> A: You can certainly toggle off the guidance rewards. We add a guidance reward to complete the tasks since, without it, the policy would not be able to finish the task. We feel like presenting a simulated sports environment without baselines rewards that work would not be well-received.
>
> > Q:  Can you not add dense rewards to encourage more human-like or realistic movement? For instance, you can penalize jerky motion, penalize joint limit violations, apply constrained RL, etc.
>
> A: Certainly! The AMP reward is precisely a dense reward to encourage human-like and realistic movement. Manually designing rewards to encourage human-like movement is an active research area [8], but it is unknown how to design rewards that lead to “human-like” behavior, which leads to many data-driven approaches in character animation.
>
> > Q: Why not also learn the holding mention, does it make the task too difficult? (similarly for tennis and table tennis, I think there could be an option to add the humanoid hand back).
>
> A: Humanoid-object interaction in simulation is an ongoing research topic in character animation, with only recent approaches demonstrating humanoid grasping objects and following diverse trajectories [9]. Thus, using the simulated humanoid with dexterous hands to learn grasping motion and play sports makes the task too difficult. Replacing the hand with the racket is a common strategy [4, 5, 10].
>
> > Q: p_t^c or c_t^b for the club position?
>
> A: p_t^b is for the ball position, and p_t^c (c_t^b is a typo) is the club position.
>
> > Q: terrain height o_t is not changing w.r.t time I guess?
>
> A: Terrain height observation o_t changes with respect to time since it represents the terrain right underneath the humanoid at time t.
>
> > Q: I suggest removing 1 x notation in eq. (1) and elsewhere for improved readability. (especially so given that you don't discuss scaling)
>
> A: The 1x notation is for the weighting of the reward component, we write it out directly here to improve the clarity and will make it clearer.
>
> > Q: is eq. (1) dense meaning that there's a reward generated at each t? (so every 1/30 seconds for a robot operated at 30Hz)
>
> A: Yes.
>
> > Q: you don't assume any air friction in eq. (3) it seems.
>
> A: Yes, the simulation we use (Isaac Gym) does not simulate air friction.
>
> > Q: what are 'qualitative results'?
>
> A: In our context, qualitative results refer to experiment results that are not presented in numbers but in videos or images.
>
> > Q: is PPO not using human demonstrations?
>
> A: No, PPO does not use human demonstrations. PPO is a popular RL algorithm that does not use any human demonstrations by itself. PPO is, however, the underlying RL algorithm used to learn policies with human demonstrations (e.g AMP, PULE).
>
> > Q: All task policies utilize three-layer MLPs with units [2048, 1024, 512]." How did you decide on this parameterization? What are the inputs to the networks? The observations of the states?
>
> A: The network size is picked following prior work in character animation. The input to the network, including observation definitions and reward definitions, are included in Appendix B. We omit them from the main text since we want to include more analysis of the results.
>
> >  Q: why only use PPO/AMP/PULSE out of many options? That wasn't clear.
>
> A: PPO/AMP/PULSE are exemplar techniques from the field of character animation. They represent the state-of-the-art animation algorithms.
>
> > Q: The main text should be as stand-alone / independent as possible from the appendix.
>
> A: We agree! We try to strike a balance between including details for each sports activity, establishing baselines, and analyzing the results. With a 10-page limit some details are omitted, and we will add the suggested details about AMP/PULSE.

---

> ### Author Response · Authors · 2024-11-25
> **Author Response to Reviewer QxJF (3/3)**
>
> > Q: "PULSE adopts a Fosbury flop technique without specific encouragement" adopts or learns / behavior emerges?
>
> A: "Adopt a technique" means choosing and using a specific method for doing something, usually a practical skill. In this context, we are referring to PULSE learning and using the Fosbury flop technique.
>
> > Q: What are these "unnatural non-human-like motions" (line 426)? Can you not engineer the rewards easily so that human-like behaviour emerges (or likelihood is increased)?
>
> A: Please refer to our anonymous site or supplementary zip (which includes all the videos) for qualitative samples. As mentioned before, engineering rewards such that human-like behavior emerges for humanoid robots without using human demonstration data would mean a major breakthrough in the community,
>
> > Q:  "w/o" vs. "w/"
>
> A: w/o: is a shorthand for ‘without’ while “w/” is for with.
>
> > Q: We find that by combining expert reward design and powerful human motion prior, one can achieve human-like behavior for solving various challenging sports" If this is one of the (and perhaps the strongest, according to the text) contributions of the paper, we'd expect it to be more carefully presented: e.g. by comparing against more RL methods / more investigations on reward design. Moreover, what is the use-case of achieving more human-like behavior? That is not discussed.
>
> A: PPO serves as the most popular and widely used RL baseline. While other RL methods may achieve better sample efficiency, they do not provide any motion prior and will lead to unnatural behavior. As for reward design analysis, see above. For use cases, we added the following paragraph in the introduction:
>
> Natural and human-like humanoid motion in sports can be used in animation, robotics motion planning, and more, where the quality of the motion is important. Human-like motion is not only visually appealing but also serves as a functional and efficient foundation for various tasks such as navigation, grasping, etc.
>
> > Q:  In fact safety may not be achieved merely through human motion priors.
>
> A: We agree; we think human motion prior can serve as the foundation for expected and safe behavior for humanoids.
>
> > Q: task observation (line 160) and goal state seem to be used interchangeably in the appendix
>
> A: Yes, task observations and goal state refer to the same thing in the text. We will clear this up.
>
>
> Typos: thank you for the suggestions! We have updated them in the updated PDF.
>
>
>
> [1] Sferrazza, Carmelo, et al. "Humanoidbench: Simulated humanoid benchmark for whole-body locomotion and manipulation." RSS 2024
>
> [2] He, Tairan, et al. "OmniH2O: Universal and Dexterous Human-to-Humanoid Whole-Body Teleoperation and Learning." CORL 2024
>
> [3] Wang et al. "Strategy and skill learning for physics-based table tennis animation." ACM SIGGRAPH 2024 Conference Papers. 2024.
>
> [4] Zhang et al. "Learning physically simulated tennis skills from broadcast videos." ACM SIGGRAPH (2023).
>
> [5] Won et al. "Control strategies for physically simulated characters performing two-player competitive sports." ACM Transactions on Graphics (TOG) 40.4 (2021): 1-11.
>
> [6] Shin, Soyong, et al. "Wham: Reconstructing world-grounded humans with accurate 3d motion." CVPR 2024
>
> [7] Li, Tianyu, et al. "Ace: Adversarial correspondence embedding for cross morphology motion retargeting from human to nonhuman characters." SIGGRAPH Asia 2023 Conference Papers. 2023.
>
> [8] Yu, Wenhao, Greg Turk, and C. Karen Liu. "Learning symmetric and low-energy locomotion." ACM Transactions on Graphics (TOG) 37.4 (2018)
>
> [9] Luo, Zhengyi, et al. "Grasping diverse objects with simulated humanoids." NeurIPS 2024.
>
> [10] D'Ambrosio, David B., et al. "Achieving human level competitive robot table tennis." arXiv e-prints (2024): arXiv-2408.

---

### Official Review · Reviewer_aCNT · 2024-11-05

**Soundness:** 1
**Presentation:** 2
**Contribution:** 2
**Rating:** 3
**Confidence:** 5

**Summary:**

This paper aims to provide a simulation environment that includes humanoid robots capable of performing a range of Olympic sports, such as long jump and tennis. However, most or all of the proposed simulation activities are ambitious and likely not feasible for current state-of-the-art humanoid robots (e.g., Unitree H1 and G1).

**Strengths:**

+ If successful, building a high-fidelity simulation of humanoid robots with diverse tasks could greatly contribute to the research community focused on humanoid robotics.

+ Demonstrating the capabilities of humanoid robots to perform Olympic games sounds fascinating.

**Weaknesses:**

- The proposed humanoid simulation is overly optimistic and ambitious, and it does not consider or justify the feasibility of deployment on real physical humanoid robots. For example, please explain how a Unitree H1 robot could perform a long jump or high jump. Can the motors provide enough torque for the robot to achieve this? The physical H1 lacks an ankle joint, so how can it perform lateral movements efficiently?

- The paper does not show validation of the simulation or methods trained in the simulation on real robots. The significant sim-to-real gap in this simulation due to simplified assumptions on physical constraints (e.g., kinematics, dynamics, and torque capabilities) makes this work less valuable to the robotics community focused on physical humanoid robots.

- In my opinion, designing a diverse set of activities is not the main bottleneck for humanoid simulation. The real challenge lies in accurately modeling a humanoid's kinematics, dynamics, torque and motion constraints, and physical interactions with the environment.

**Questions:**

See above.

---

> ### Author Response · Authors · 2024-11-25
> **Author Reponse to Reviewer aCNT**
>
> Thank you for your review! Here we address your concerns in the hope that the reviewer raises the score.
>
> **Justify the feasibility of deployment on real physical humanoid robots.**
>
> We use the same simulator, humanoid models, PD controllers and torque limits, and simulation parameters as contemporary sim-to-real works using the H1 humanoid [1, 2, 3]. Our environments are lightweight, so adding the domain randomization and control delays needed for sim-to-real transfer would be straightforward. Indeed, the high jump and long jump have not been shown to be possible using the real H1 humanoid, but that’s precisely why we would like to explore the task in simulation first and push the boundary of the humanoids. We believe that first trying to solve these tasks in simulation and then iteratively aligning the simulation with the real-world deployment is the way to achieve these feats with the real humanoid. As for motor strength, the Unitree H1 humanoid can backflip [4], and parkour [5], which demonstrates the motor strength of the H1 humanoid has the potential to achieve some of the sports we designed.
>
> Exploring these tasks in simulation first paves the way for us to build toward the behavior we want on real humanoids and offers us a chance to find the proper motion prior to learning the algorithm. In our paper, we do not claim any real-world deployment capabilities.
>
> **The real challenge lies in accurately modeling a humanoid's kinematics, dynamics, torque and motion constraints, and physical interactions with the environment.**
>
> We agree that accurately modeling the humanoid’s kinematics and dynamics in simulation is crucial for deployment. We use the same simulation parameters and humanoid models that have been shown for sim-to-real transfer for this purpose. On the other hand, we also believe that the motor skills found in human data are the key to unlocking the potential of humanoids, and human motion can serve as a strong foundation for humanoid motion. The result shown in this paper supports this finding, and our provided video-to-human-pose and retargeting pipeline can serve as the foundation to provide humanoids with such capabilities.
>
> **Contributions to the Robotics Community**
>
> Our main contribution is designing sports environments for **simulated** humanoids which serve as a basis for modeling humanoid behaviors. As sim-to-real continues to work, we believe there is ample synergy to be explored between the graphics community and the robotics community, where techniques in character animation can be applied to real robot control [6]. Humanoid Olympics can help the **animation and robotics communities** develop human-like and performant controllers by providing a unified and competitive sports benchmark.
>
>
> [1] He, Tairan, et al. "OmniH2O: Universal and Dexterous Human-to-Humanoid Whole-Body Teleoperation and Learning." CORL 2024
>
> [2] Cheng, Xuxin, et al. "Expressive whole-body control for humanoid robots." RSS 2024
>
> [3] Fu, Zipeng, et al. "HumanPlus: Humanoid Shadowing and Imitation from Humans." CORL 2024.
>
> [4] https://www.youtube.com/watch?v=V1LyWsiTgms
>
> [5] Zhuang, et al.  "Humanoid Parkour Learning."
>
> [6] Serifi, Agon, et al. "Vmp: Versatile motion priors for robustly tracking motion on physical characters." Computer Graphics Forum. 2024.

---

### Official Review · Reviewer_xKNz · 2024-11-06

**Soundness:** 2
**Presentation:** 3
**Contribution:** 2
**Rating:** 5
**Confidence:** 4

**Summary:**

This work presents a collection of environment settings in Isaac Gym and the benchmark for single-person sports and multi-person sports using several reinforcement learning algorithms. Human demonstration data could be obtained from videos using some existing motion reconstruction pipeline.

**Strengths:**

The tasks in this work are about sport activities. These tasks are not easy and a variety of tasks are included ranging from simple actions to complex competition scenarios. It also shows processed human demonstration data from videos are helpful for the learning tasks and very detailed benchmark results are provided.

**Weaknesses:**

This work leverages several existing components to build a framework for training simulated humanoids. Although the tasks are interesting, the overall contribution is kind of marginal because most of the work is about implementation and benchmark of existing algorithms.

**Questions:**

1. What are the challenges to set up the training environment for sports in Issac Gym compared with other tasks? This could be a good motivation and make the work not just about implementation.
2. The reward functions are carefully designed for these tasks which is also claimed to be very important for good simulation results. I am wondering how much better these reward functions could be compared with simple reward functions?

---

> ### Author Response · Authors · 2024-11-25
> **Author Reponse to Reviewer xKNz**
>
> The authors thank the reviewer for the thoughtful reviews! Here we try to address your concerns in the hope that the reviewer raises the score.
>
> **Most of the work is about the implementation and benchmarking of existing algorithms**
>
> Exactly! We are aiming for the datasets & benchmark as the primary area and focus on building simulated sports environments for humanoids. We test existing algorithms to not only test how far we have come as a community but also provide novel results by designing diverse environments.
>
> **Challenges in setting up training environments for sports**
>
> Building a collection of simulated sports environments that supports a range of standardized humanoid embodiment and training pipelines is challenging, as it requires expert knowledge in humanoid control, reinforcement learning (RL), and physics simulation. Each sport requires setting up the assets for playing the sport (scene, humanoid, and sometimes objects), initialization conditions, rewards for training the agents, and termination conditions. For sports like ping-pong [1], tennis [2], boxing, and fencing [3], there have been individual full papers that work on them, while we provide a unified training & testing ground for all sports centered around using common humanoids for each one of the sports.
>
>
> **Importance of the Reward Functions**
>
> The expert reward function we designed provides the human-like behavior we can see from our qualitative examples. With simple rewards based on the sports' objective, many tasks would not be learned (e.g., the javelin won’t be thrown, or the ball will not be hit towards the goalie), as the objective can be very sparse. The reward function serves as a guide to navigating the sparse signal provided by the final task objective.
>
>
> [1] Wang et al. "Strategy and skill learning for physics-based table tennis animation." ACM SIGGRAPH 2024 Conference Papers. 2024.
>
> [2] Zhang et al. "Learning physically simulated tennis skills from broadcast videos." ACM SIGGRAPH (2023).
>
> [3] Won et al. "Control strategies for physically simulated characters performing two-player competitive sports." ACM Transactions on Graphics (TOG) 40.4 (2021): 1-11.

---

### Author Response · Authors · 2024-11-25
**Rebuttal Revision**

We thank reviewers for the reviews and made the following revision:

- Fixed typos.
- Added discussion for AMP & PULSE.
- Added motivation for human-like motion

---

### Meta-Review · Area_Chair_KpCB · 2024-12-23

**Metareview:**

The paper proposes a series of simulation-based environments for the training and benchmarking of humanoid robots in the context of single- and multi-person sports tasks. The paper evaluates policies trained in these environments via reinforcement learning (RL) using dense rewards proposed in the paper, as well as policies trained on human demonstration data.

The problem of learning control policies for humanoid robots continues to garner significant attention. Because of this, several reviewers find that with the inclusion of a high-fidelity simulator for humanoid robots along with what is an interesting set of environments, the work could significantly benefit the research community. However, the reviewers largely agree that the contributions of the paper in its current form are marginal. As one reviewer points out, it is not apparent that the main bottleneck to the realization of effective control policies for humanoid robots is not the lack of a sufficiently diverse set of tasks. An updated version of the paper that significantly expanded the focus on learning, e.g., by evaluating the performance of different RL algorithms, the effects of reward design, sim-to-real performance, etc., would provide a valuable contribution to researchers working on humanoid control.

**Additional Comments On Reviewer Discussion:**

There was consensus among the reviewers in terms of the paper's strengths and weaknesses.

---

### Decision · Program_Chairs · 2025-01-22

Reject